# Stepwise recombination suppression around the mating-type locus in an ascomycete fungus with self-fertile spores

Nina Vittorelli[1,2,3‡], Ricardo C. Rodríguez de la Vega[1‡], Alodie Snirc[1], Emilie Levert[1,2], Valérie Gautier[2], Christophe Lalanne[2], Elsa De Filippo[1,2], Pierre Gladieux[4], Sonia Guillou[4], Yu Zhang[5], Sravanthi Tejomurthula[5], Igor V. Grigoriev[5,6], Robert Debuchy[7], Philippe Silar[2‡], Tatiana Giraud[1‡], Fanny E. Hartmann[1‡]*

1 Ecologie Systematique et Evolution, CNRS, Université Paris-Saclay, AgroParisTech, Gif-sur-Yvette, France, 2 Laboratoire Interdisciplinaire des Energies de Demain, Université Paris Cité, Paris, France, 3 Département de Biologie, École Normale Supérieure, PSL Université Paris, Paris, France, 4 PHIM Plant Health Institute, Univ Montpellier, INRAE, CIRAD, Institut Agro, IRD, Montpellier, France, 5 U.S. Department of Energy Joint Genome Institute, Lawrence Berkeley National Laboratory, Berkeley, California, United States of America, 6 Department of Plant and Microbial Biology, University of California Berkeley, Berkeley, California, United States of America, 7 Université Paris-Saclay, CEA, CNRS, Institute for Integrative Biology of the Cell (I2BC), Gif-sur-Yvette, France

‡ NV and RCR share first authorship on this work. PS, TG and FET are joint senior authors on this work.
* fanny.hartmann@universite-paris-saclay.fr

**Data Availability Statement:** All genomic data produced by the JGI are available from the JGI MycoCosm website (https://mycocosm.jgi.doe. gov/Podtet1/Podtet1.home.html, last accessed

## Abstract

Recombination is often suppressed at sex-determining loci in plants and animals, and at self-incompatibility or mating-type loci in plants and fungi. In fungal ascomycetes, recombination suppression around the mating-type locus is associated with pseudo-homothallism, *i. e.* the production of self-fertile dikaryotic sexual spores carrying the two opposite mating types. This has been well studied in two species complexes from different families of *Sordariales*: *Podospora anserina* and *Neurospora tetrasperma*. However, it is unclear whether this intriguing association holds in other species. We show here that *Schizothecium tetrasporum*, a fungus from a third family in the order *Sordariales*, also produces mostly self-fertile dikaryotic spores carrying the two opposite mating types. This was due to a high frequency of second meiotic division segregation at the mating-type locus, indicating the occurrence of a single and systematic crossing-over event between the mating-type locus and the centromere, as in *P. anserina*. The mating-type locus has the typical *Sordariales* organization, plus a *MAT1-1-1* pseudogene in the *MAT1-2* haplotype. High-quality genome assemblies of opposite mating types and segregation analyses revealed a suppression of recombination in a region of 1.47 Mb around the mating-type locus. We detected three evolutionary strata, indicating a stepwise extension of recombination suppression. The three strata displayed no rearrangement or transposable element accumulation but gene losses and gene disruptions were present, and precisely at the strata margins. Our findings indicate a convergent evolution of self-fertile dikaryotic sexual spores across multiple ascomycete fungi. The particular pattern of meiotic segregation at the mating-type locus was associated with recombination suppression around this locus, that had extended stepwise. This association between pseudo-homothallism and recombination suppression across lineages and

February 5, 2022). The raw Illumina reads produced at Genoscope, the Nanopore reads and the long read-based assemblies were deposited in GenBank under the BioProject accession number PRJNA882797 (BioSample ID: SAMN30962396 for CBS815.71-sp3 and BioSample ID: SAMN30962397 for CBS815.71-sp6). Accession numbers of the raw Illumina reads are SRX17718707 for CBS815.71-sp3 and SRX17718708 for CBS815.71-sp6. Accession numbers of the Nanopore reads are SRX17801082 for CBS815.71-sp3 and SRX17801083 for CBS815.71-sp6. Accession numbers of the long read-based assemblies are JAQKAE000000000 for CBS815.71-sp3 and JAQKAD000000000 for CBS815.71-sp6.

**Funding:** This work was supported by the Louis D. Foundation award (Institut de France) and EvolSexChrom ERC advanced grant #832352 (H2020 European Research Council) to T.G., the ANR PIA grant # ANR-20-IDEES-0002 to F.E.H., the DOE-JGI CSP (Joint Genome Institute) grant #504394 to P.G. and P.S, the work (proposal 10.46936/10.25585/60001199) conducted by the US Department of Energy Joint Genome Institute (JGI: https://ror.org/04xm1d337), a DOE Office of Science User Facility, and supported by the Office of Science of the US Department of Energy under contract no. DE-AC02-05CH11231. The funders had no role in study design, data collection and analysis, decision to publish, or preparation of the manuscript.

**Competing interests:** The authors have no competing interests.

the presence of gene disruption at the strata limits are consistent with a recently proposed mechanism of sheltering deleterious alleles to explain stepwise recombination suppression.

## Author summary

Recombination allows faster adaptation and the purging of deleterious mutations but is often paradoxically lacking in sex chromosomes. It has been recently recognized that recombination can also be suppressed on fungal mating-type chromosomes, but the evolutionary explanation and the proximal mechanism of this phenomenon remain unclear. By studying here the sexual biology of a poorly studied mold living in rabbit dung, we reveal a striking convergence in three distant fungal lineages of an independently evolved association between the production of self-fertile sexual spores (carrying two nuclei with opposite mating types), a particular segregation of the mating-type locus and stepwise recombination suppression on mating-type chromosomes. Such a convergent association suggests causal relationships and will contribute to unveil the evolutionary causes of recombination suppression.

## Introduction

In eukaryotes, recombination is often considered to be a major long-term evolutionary advantage. Indeed, it prevents the accumulation of deleterious mutations [1] and facilitates the combination of multiple advantageous alleles in the same genome [2]. However, convergent recombination suppression events have been observed at sex-determining loci in plants and animals, and at self-incompatibility or mating-type loci in plants and fungi [3]. It has been suggested that this recombination suppression links lock-and-key genes together, enabling self-incompatibility systems to function correctly [4–6]. Surprisingly, recombination suppression has often been found to extend distally, in a stepwise manner, from sex and mating-type loci, generating evolutionary strata of differentiation between haplotypes [4, 7–9]. The evolutionary causes of this stepwise recombination suppression beyond mating-compatibility loci remain a matter of debate [4, 9–11]. Sexual antagonism was long thought to be the main driving force generating evolutionary strata on sex chromosomes, through selection for the linkage to the sex-determining locus of genes with alleles beneficial in only one sex [11]. However, it has been difficult to obtain definitive evidence supporting this hypothesis, and evolutionary strata have also been reported in fungi without sexual antagonism or other types of antagonism between mating types [4, 8, 9, 12, 13]. Alternative hypotheses have been developed [4, 10], based on dosage compensation [14], the spread of transposable elements [15] or the selection of a non-recombining fragment with a lower mutation load than average, that eventually reaches maximal frequency thanks to the sheltering of its few deleterious mutations by a nearby mating-type or sex-determining locus [16].

Fungi are useful models for studies of the evolution of recombination suppression: they have many assets for evolutionary genomics studies [17] and several convergent recombination suppression events around mating-type loci have been observed in this group [4, 8, 18]. Fungi have highly diverse modes of reproduction, mating systems (selfing versus outcrossing) and breeding systems (mating compatibility), with considerable differences sometimes reported between closely related species [19]. Among all breeding systems, heterothallism was the most investigated and seems to display the most conserved scheme in fungi [20]. In

heterothallic fungi, mating can occur only between haploid cells carrying different alleles at the mating-type locus. The term "heterothallism" was initially coined after observations of *in vitro* crosses, in which mating could occur only between different thalli (*i.e.* mycelia or hyphal networks), each arising from a single spore. We now know that this heterothallism was due to the thalli mostly being haploid and carrying different alleles at the mating-type locus. Other fungi were described as "homothallic" because mating within a single thallus was possible. Indeed, mating in these fungi does not need to occur between cells of different mating types. However, some fungi capable of displaying mating within a single thallus *in vitro* were later found to carry two types of nuclei, of different mating types, in each cell. They were therefore called pseudo-homothallic.

The pseudo-homothallic breeding system has been studied in detail in two species from different families of the order *Sordariales* (*Ascomycota*): *Neurospora tetrasperma* (*Sordariaceae)*, which is typically found as orange blooms on scorched vegetation at burnt sites after a fire, and *Podospora anserina* (*Podosporaceae*), which is mostly found in herbivore dung. Sexual spore formation is similar in these two species. Karyogamy occurs between two haploid nuclei of different mating types, followed by meiosis, and then the post-meiotic mitosis of each meiosis product. The ascus produced therefore contains eight haploid nuclei, which are packaged in pairs to form four sexual spores (ascospores) [21, 22]. The resulting ascospores are thus dikaryotic, *i.e.* each contains two haploid nuclei (Fig 1). A small fraction of asci (0.1% for *N. tetrasperma* and 1.0% in *P. anserina*) contains five ascospores, with one dikaryotic spore replaced by a pair of monokaryotic ascospores (*i.e.* ascospores carrying a single haploid nucleus), recognizable in *P. anserina* by their smaller size. In both species, the dikaryotic spores almost always contain nuclei of opposite mating types, and are therefore referred to as heterokaryotic ascospores [23, 24], each of which germinates into a self-fertile mycelium [25]. The particular pattern of segregation of the mating-type locus leads to the generation of nuclei of different mating types in close physical proximity, which can be compacted into a single ascospore (Fig 1). In *N. tetrasperma*, the mating-type locus segregates during the first meiotic division, due to its linkage to the centromere. During meiosis II, the spindles are parallel, enabling two nuclei with opposite mating types to be present at each pole of the ascus. Following the post-meiotic mitosis, each ascospore is delineated around two non-sister nuclei, resulting in the compaction of opposite mating types in the same cell (Fig 1A). In *P. anserina*, the mating-type locus segregates during the second meiotic division, due to the occurrence of a single and systematic crossing-over (CO) event between the centromere and the mating-type locus. During meiosis II, the spindles are aligned and it is the peculiar pattern of ascospore delimitation following the post-meiotic mitosis, encompassing sister nuclei, that leads to the same final nuclear distribution as in *N. tetrasperma* (Fig 1B).

In these two pseudo-homothallic species, there are large non-recombining regions around the mating-type locus. In *N. tetrasperma*, the 7 Mb region without recombination encompasses the mating-type locus and the centromere [26]. In *P. anserina*, a 0.8 Mb region devoid of recombination has been identified around the mating-type locus in the reference strain [27]. Each of these two species belongs to a complex of cryptic species [26, 28, 29], which also all harbor a region of recombination suppression around their mating-type locus; however the size of the non-recombining region differs between species within each species complex [26, 30, 31]. In both *N. tetrasperma* and *P. pseudocomata*, a cryptic species from the *P. anserina* species complex [28], the suppression of recombination has extended stepwise, generating evolutionary strata [4, 26, 30, 31]. It is intriguing that the presence of non-recombining regions around fungal mating-type loci and their stepwise extension seem to be associated with pseudo-homothallism in ascomycetes [4]. There is indeed no recombination suppression around the mating-type locus reported so far in true heterothallic ascomycetes, as seen in

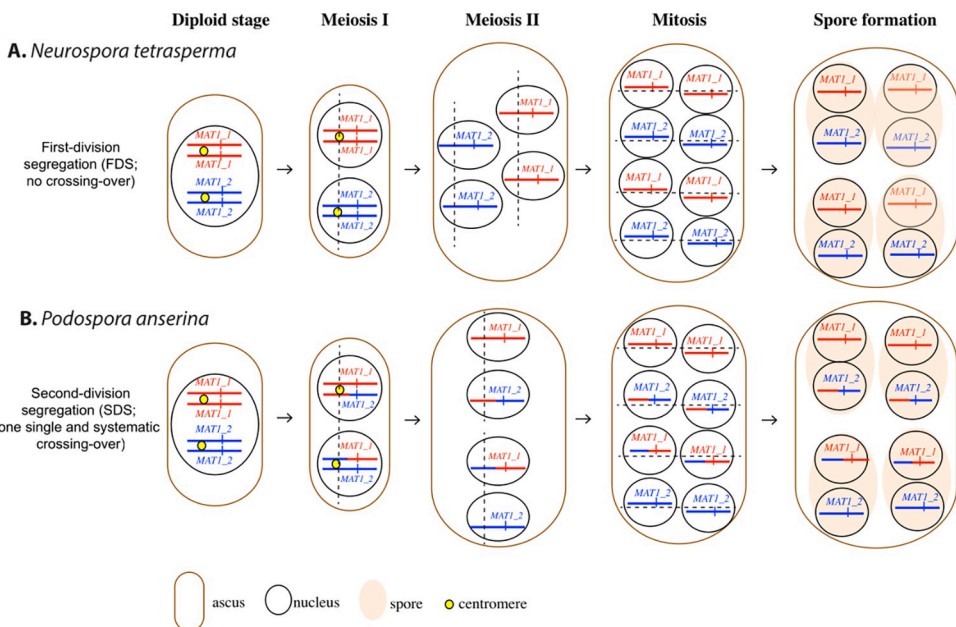

**Fig 1. Mating-type segregation and nuclei compaction in spores during ascus development in the pseudo-homothallic fungi** *Neurospora tetrasperma* **(A) and** *Podospora anserina* **(B) (***Sordariales***,** *Ascomycota***).** In each panel, five stages are represented from left to right: the diploid cell before meiosis with a nucleus heterozygous at the mating-type locus; the first division of meiosis (meiosis I); the second division of meiosis (meiosis II); the post-meiotic mitoses (PMM); and the spore formation with the compaction of non-sister nuclei in *N. tetrasperma* and sister nuclei in *P. anserina*. Only the chromosomes carrying the mating-type locus are represented, with their centromeres. The opposite *MAT1-1* and *MAT1-2* alleles at the mating-type locus are represented by red and blue vertical bars, respectively. The mating-type locus segregate at meiosis I in *N. tetrasperma* and at meiosis II in *P. anserina*. During meiosis II, spindles are parallel in *N. tetrasperma* and aligned in *P. anserina*. The standard nomenclature of mating-type alleles is used, with *MAT1-2* corresponding to the *mat+* and *mat_a* allele and *MAT1-1* to the *mat-* and *mat_A* allele in *P. anserina* and *N. tetrasperma*, respectively. Centromeres are represented by yellow circles at the diploid and meiosis I stages. Chromosomal arms are colored in red or blue according to the parental origin of the chromosomal fragment. Spindle orientations in meiosis and mitoses are indicated by dotted line. The figure was adapted from Ref (4).

available genetic maps [32–41]. An exception is *Cryphonectria parasitica* [42], in which dikaryons are nevertheless relatively frequently isolated in nature [43–45]. One hypothesis that could account for this association between pseudo-homothallism and recombination suppression around the mating-type locus in ascomycetes is that most strains are heterokaryotic at the mating-type locus for most of their life cycle in pseudo-homothallic ascomycetes, while heterothallic ascomycetes have a dominant haploid (or monokaryotic) phase. Being heterokaryotic at the mating-type locus could allow recombination suppression to be selected as a means of protecting a genomic region containing fewer deleterious mutations than the population average while sheltering its few recessive deleterious alleles by linking them to a permanently heterozygous locus [16]. Indeed, non-recombinant haplotypes with fewer recessive deleterious mutations than the population average should be selected for when rare, but if they increase in frequency, they will start to occur as homozygotes, exposing their recessive load, unless they capture a permanently heterozygous allele. In such cases, they continue to be selected and can rapidly reach their maximal frequency, *i.e.* becoming fully associated with the heterozygous allele close to which they appeared [16]. This mechanism only works with an extended diploid (or dikaryotic) phase and with a bi-allelic mating-type locus. Indeed, an extended haploid (or monokaryotic) phase exposes deleterious alleles to selection and the existence of multiple mating-type alleles allows losing an allele associated with an evolutionary stratum after it has

accumulated too many deleterious mutations following recombination suppression [16, 46]. Automixis (intra-tetrad mating) facilitates the conditions for an extension of recombination suppression in selfing organisms [47, 48]. The extended dikaryotic phase in pseudo-homothallic ascomycetes, their bi-allelic mating-type locus and their automictic mating system should therefore foster a stepwise extension of recombination suppression around the mating-type locus by this mechanism [16]. Basidiomycete fungi are dikaryotic across most of their life cycle, but they often have multi-allelic mating-type loci [49], allowing the loss of those with lower fitness. A few cases of recombination suppression near mating-type loci have been reported in this group, in particular in bipolar basidiomycetes, but only automictic basidiomycete lineages with a bi-allelic mating-type locus have extended recombination suppression [8, 9, 18, 50], which is consistent with the deleterious mutation sheltering model. Studies of additional pseudo-homothallic ascomycete fungi are required, to assess the strength of the association between pseudo-homothallism and stepwise recombination suppression around the mating-type locus.

We therefore studied *Schizothecium tetrasporum* (previously known as *Podospora tetraspora* or *P. minuta* var. *tetraspora)*, a third pseudo-homothallic species from order *Sordariales* [21, 25, 51, 52]. *Schizothecium tetrasporum* belongs to the family *Schizotheciaceae*, while *P. anserina* belongs to the *Podosporaceae* and *N. tetrasperma* to the *Sordariaceae* [53]. The phylogenetic relationships between these three families are not well established, but they probably diverged over 150 million years ago [54]. For example, the phylogenetic distance between *P. anserina* and *N. crassa*, a member of the *Sordariaceae* (like *N. tetrasperma)*, corresponds roughly to that between humans and fishes [55]. Phylogenetically, *Schizothecium tetrasporum* is roughly equidistant from these two species [53]. However, it displays striking cytological similarities to *P. anserina*, particularly in terms of dikaryotic spore production: in the FGSC7436 strain, 98% of asci were reported to produce four dikaryotic spores, with the spindles aligned in the ascus during meiosis II [25]. The substantial phylogenetic distance between these pseudo-homothallic species, that are each closely related to true heterothallic and homothallic species, indicates that pseudo-homothallism has evolved independently in these three lineages. The previous study on *S. tetrasporum* [25] did not investigate whether the two nuclei of a dikaryotic spore contained opposite mating types and did not study the mating-type locus or recombination suppression.

We identified media suitable for reproducible and fast growth, and media for sexual reproduction in *S. tetrasporum*. We studied the dikaryotic strain CBS815.71, estimating the frequency of second-division segregation of the mating-type locus at 99.5%. This indicates the existence of a single and systematic CO event between the mating-type locus and the centromere. We also built high-quality genome assemblies of two homokaryons (*i.e.* with only one nucleus type), with opposite mating types, making it possible: i) to elucidate the genetic organization of the mating-type locus, and ii) to detect a 1.47 Mb region without recombination around the mating-type locus, through analysis of differentiation between the mating-type chromosomes. We observed three evolutionary strata, with significantly different levels of differentiation between mating-type chromosomes, suggesting a stepwise extension of recombination suppression. Segregation analyses confirmed the suppression of recombination. We found signs of deleterious mutations in terms of gene losses and gene disruptions precisely at the margins of evolutionary strata, but no increase in transposable element content, non-synonymous substitutions or rearrangements. Our findings indicate that the convergent evolution of an extremely high frequency of self-fertile dikaryotic spores together with recombination suppression around the mating-type locus has occurred in multiple pseudo-homothallic fungal species.

## Results

### High frequency of self-fertile spores in *Schizothecium tetrasporum*

The study of a single strain of *Schizothecium tetrasporum* (Fig 2) suggested that this species was pseudo-homothallic and produced self-fertile heterokaryotic ascospores similarly to *P. anserina* [21, 25], *i.e.* with most asci containing four dikaryotic spores, probably carrying both mating types. We studied the breeding system of *S. tetrasporum* further, by testing different media and identifying those optimal for reproducible and fast growth, and those optimal for sexual reproduction (see Material & Method section). Raju & Perkins found that the FGSC7436 strain produced 2% five-spore asci. In this study, we found that 10% of the asci produced by the CBS815.71 strain had five spores. We also observed, in rare frequencies, other kinds of asci, including six-spore asci, and some with "extra-large" ascospores, probably encompassing three nuclei, rather than the two usually grouped together during ascospore delimitation (Fig 2). The resulting mycelium produced numerous asexual single-cell propagules, for which we were unable to find appropriate conditions for germination and development. Therefore, these cells probably corresponded to spermatia rather than conidia, *i.e.* they are sexual cells acting as fertilizing elements and are unable to germinate, in contrast to asexual spores. *Podospora anserina* also produces spermatia, which function exclusively in the fertilization of the female structures, the protoperithecia [56]. We generated an F1 progeny from the CBS815.71 dikaryotic strain, and isolated two homokaryotic offspring of opposite mating types, referred to hereafter as CBS815.71-sp3 and CBS815.71-sp6. The mating types of these strains were named according to the standard nomenclature [57]: *MAT1-1* for CBS815.71-sp6 and *MAT1-2* for CBS815.71-sp3.

Previous cytological observations indicated second-division segregation (SDS) at the mating-type locus in *S. tetrasporum* [25] and we assessed here its frequency (Fig 3), by analyzing

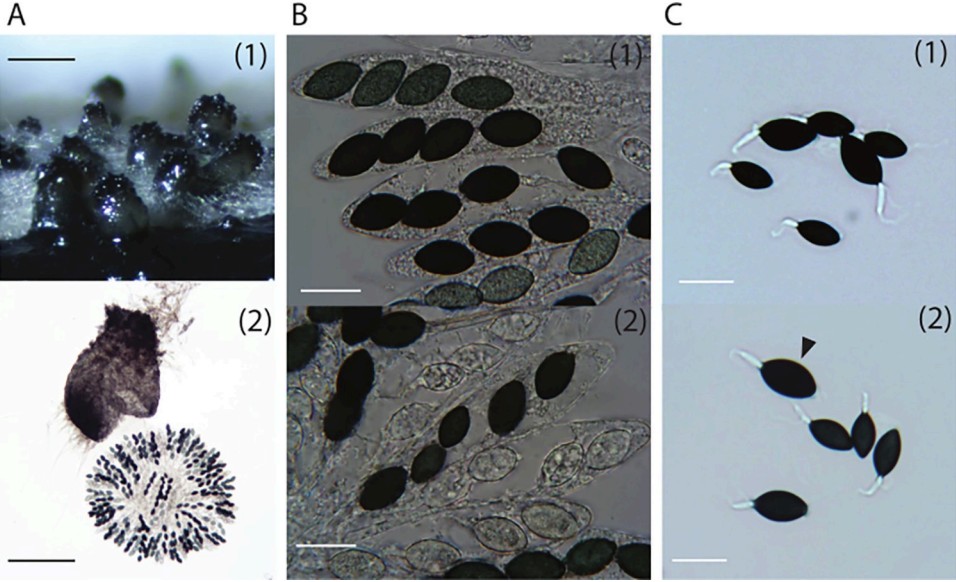

**Fig 2. Sexual reproductive structures and ascospores of *Schizothecium tetrasporum* CBS815.71. A.** Perithecia on *Miscanthus* [1] and squashed perithecia showing the rosette of asci [2]; bars = 200 μm. **B.** Enlargements of asci within the rosettes [1] and a five-spore ascus with three dikaryotic ascospores and two monokaryotic ascospores [2]; bars = 20 μm. **C.** Other abnormal asci observed in CBS815.71: [1] a six-spore ascus with four monokaryotic ascospores (small) and two dikaryotic ones (large); [2] a five-spore ascus with three monokaryotic ascospores (the three small spores in the middle), one dikaryotic (the large ascospore at the bottom) and another, probably trikaryotic, ascospore (arrowhead showing the very large ascospore); bars = 20 μm.

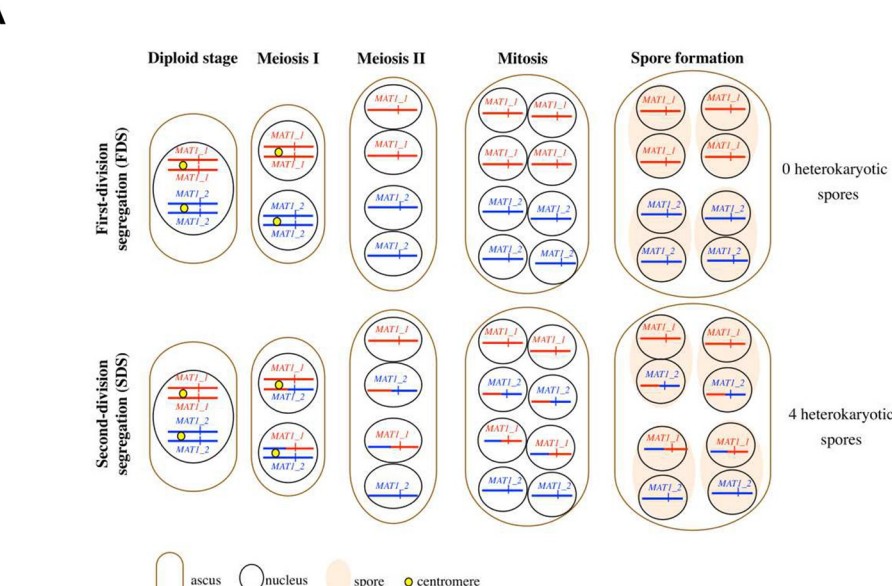

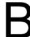

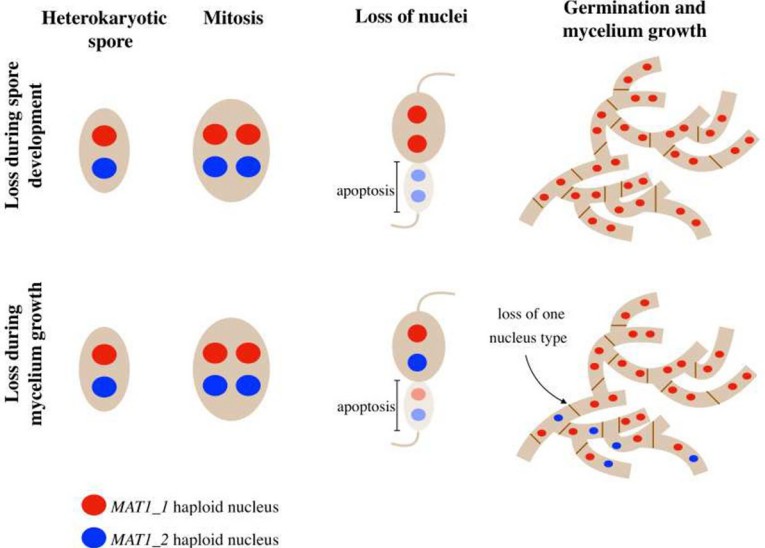

**Fig 3. Tested scenarios of segregation of the mating-type locus during meiosis and theoretical ascus composition and hypothetical scenarios for the loss of one mating type in thalli of *Schizothecium tetrasporum*. A.** Theoretical composition of an ascus according to meiotic segregation pattern at the mating-type locus: if alleles at the mating-type locus (represented as a red or blue vertical bar) segregate during the first meiotic division (FDS, top), two *MAT-1-1* spores and two *MAT-1-2* spores are formed; conversely, if alleles at the mating-type locus segregate during the second meiotic division (SDS, bottom), four heterokaryotic spores carrying both mating types are formed. Only the chromosomes carrying the mating-type locus are represented, with their centromeres. The *MAT-1-1* and *MAT1-2* alleles at the mating-type locus are represented by red and blue vertical bars, respectively. Centromeres are represented by yellow circles at the diploid and meiosis I stages. Chromosomal arms are colored in red or blue according to the parental origin of the chromosomal fragment. **B.** Two hypothetical scenarios for the loss of one mating type despite SDS at the mating-type locus. SDS at the mating-type locus generates a spore carrying two nuclei of opposite mating types, but one nucleus may be lost during spore development (top) when half the nuclei gather in a region of the spore that subsequently undergoes degeneration, or during mycelium growth (bottom).

the progeny of a cross between CBS815.71-sp3 and CBS815.71-sp6. We collected 190 four-spore asci (each ascus containing the products of a given meiosis) and isolated large (dikaryotic) ascospores. Mating types were determined in the mycelium generated from germinating ascospores, by PCR with two primer pairs, each specific to one mating-type allele and yielding amplicons of different sizes. We initially genotyped only one ascospore per ascus, as this is usually sufficient to differentiate between two extreme cases: first-division segregation at meiosis (FDS) and second-division segregation (SDS). FDS yields four-spore asci containing two ascospores of a single mating type and another two ascospores of the opposite mating type, whereas SDS results in four ascospores, all containing both mating types (Fig 3A). We found that 87% (166/190) of the thalli resulting from ascospore germination carried both mating types, indicating that they resulted from SDS, and 13% (24/190) carried a single mating type, being therefore either haploid or with two nuclei of the same mating type. Indeed, thalli carrying a single mating type may result not only from FDS, but also from the loss of a nucleus during growth in thalli resulting from SDS (Fig 3B). We differentiated between these two situations by determining the alleles at the mating-type locus in thalli resulting from the germination of the three remaining spores from the 24 four-spore asci in which we found a single mating type in the first genotyped ascospore. Only one of the 24 asci analyzed proved to be a true FDS ascus (*i.e.* with two *MAT1-1* and two *MAT1-2* ascospores). The other 23 asci contained at least one ascospore carrying both mating types, indicating that the homokaryotic thalli growing from their other ascospores resulted from the loss of nuclei during growth after SDS in asci (Fig 3B). We estimated the overall frequency of SDS for mating type at 99.5% (189/190) in the CBS815.71 strain.

We detected a substantial number of homokaryotic thalli originating from SDS asci (23 thalli from 189 germinating heterokaryotic ascospores), suggesting that the loss of nuclei corresponding to one mating type in heterokaryotic thalli may occur at a non-negligible frequency in the CBS815.71 strain. Raju and Perkins [25] showed that the two nuclei present in developing ascospores undergo mitosis, which is rapidly followed by the migration of two of the four daughter nuclei into the future pedicel cell, which are presumed to abort, or at least to fail to contribute to the nuclear composition of the resulting culture. It is therefore possible that homokaryotic thalli result from the migration of two nuclei carrying the same mating type into the future pedicel (leaving two nuclei of the same mating type in the maturing large ascospore cell), rather than the migration of two nuclei carrying opposite mating types (leaving two nuclei of opposite mating types in the maturing large ascospore cell; Fig 3B). It is difficult to test this hypothesis, because there is no easy way to follow the migration of nuclei during ascospore maturation. Alternatively, *MAT1-1/MAT1-2* heterokaryons may sometimes lose one type of nucleus during vegetative growth.

We tested this second hypothesis by determining mating type in several sections of heterokaryotic thalli. A heterokaryon was generated by vegetative fusion (*i.e.* by mixing identical amounts of ground mycelia of the two homokaryotic strains of opposite mating types, CBS815.71-sp3 and CBS815-sp6). This procedure of mixed homokaryons chopped in very fine grain leads to the generation of a heterokaryotic mycelium through multiple vegetative fusions [58] and nuclear movements in the mycelial interconnected network of cells [59]; no sexual reproduction can occur under these growth conditions. The resulting heterokaryotic mycelium was grown on V8 medium and incubated until the colony had reached 4 cm in diameter. It was then sliced into 70 pieces, each with a volume of about 1 mm$^3$. PCR analyses demonstrated that only 31 of these 70 pieces carried both mating types, 29 contained only *MAT1-2* and 10 contained only *MAT1-1*. In a parallel experiment on another thallus of the same heterokaryon, the 70 pieces of mycelium were used to inoculate fresh M2 medium, and regenerating thalli were allowed to grow to a diameter of 1 cm. A randomly selected 1 mm$^3$ piece of each of

these thalli was sampled at the edge of growing colonies and genotyped by PCR for mating type. We found that 34 of the 70 regenerated thalli carried both mating types, whereas 25 carried only *MAT1-2* and 11 carried only *MAT1-1*. These results indicate that one of the two nuclei can be lost during vegetative growth, and they confirm the high rate of SDS in *S. tetrasporum*, yielding 99.5% of asci with all four ascospores initially containing both mating types, but with the occasional loss of one nucleus during vegetative growth. The *MAT1-1* mating type was preferentially lost in both experimental set-ups (chi$^2$ = 9.25, $p<0.005$ for the first experiment and chi$^2$ = 4.44, $p<0.025$ in the second experiment). Understanding further the sorting of *MAT1-1* vs *MAT1-2* nuclei as homokaryons will require a dynamic analysis of their movements in apical hyphae, which will need the setup of a transformation system that will enable the labelling of nuclei with different markers.

We investigated whether this loss of nuclei could be due to the relative fitness of the two types of nuclei, by investigating whether the two CBS815.71 homokaryons, carrying either the *MAT1-1* or *MAT1-2* mating type, displayed differences in growth rate or biomass production, and comparing them with the *MAT1-1/MAT1-2* heterokaryon. We measured growth rates on three different media and determined the dry weight obtained after growth on M2 medium for both homokaryons and the heterokaryon of the CBS815.71 strain (S1A Table). No significant differences in growth or biomass were found, under these conditions, between the two homokaryons or between the homokaryons and the heterokaryon, the only significant effect being that of culture medium on growth rate (ANOVA, S1B Table). The growth rate of the *MAT1-1* homokaryon, corresponding to the mating type most often lost during growth, was slightly lower, although not significantly so (S1A Table).

## Reference genome assembly for *Schizothecium tetrasporum*

Using both Illumina and Oxford Nanopore MinION technology, we sequenced the genomes of the two homokaryotic strains of opposite mating types, CBS815.71-sp3 and CBS815.71-sp6. We built a *de novo* long read-based genome assembly for each, the first such assemblies to have been generated for *S. tetrasporum*. For CBS815.71-sp3, we also had an Illumina-based assembly available that had been obtained before the long-read assemblies. BLAST searches and analyses of Illumina read coverage against the CBS815.71-sp3 long read-based assembly suggested the presence of bacterial contamination in the Nanopore data, but not in the Illumina data. We discarded scaffolds with high similarity scores to bacterial genomes in BLAST analyses for further analyses. The resulting long read-based genome assemblies for CBS815.71-sp3 and CBS815.71-sp6 contained 16 and 23 contigs and were 32.88 Mb and 32.83 Mb long, respectively. We found 96% of the 3,817 highly conserved orthologs in Sordariomycota from BUSCO analyses on the assemblies (S2 Table). In the CBS815.71-sp3 assembly, seven contigs were more than 3.5 Mb long and contained all the conserved BUSCO orthologs. One contig corresponded to the mitochondrial genome and the other eight contigs were less than 70 kb long, which is consistent with the expected number of chromosomes in *Sordariales*, an order in which several species have seven chromosomes [21, 60]. Telomeric repeats were present at both ends of contigs 1, 2, 4 and 5 in the CBS815.71-sp3 assembly, indicating that these contigs represent four entirely assembled chromosomes (Fig 3A). For CBS815.71-sp3, we compared the long read-based assembly obtained with the more fragmented Illumina-based assembly and found high levels of similarity and synteny (S1 Fig). We used the long read-based assembly of CBS815.71-sp3 for further analyses. We found a high overall level of collinearity between the two homokaryotic long read-based genome assemblies for CBS815.71-sp3 and CBS815.71-sp6 (Fig 3B). We found over 98% of nucleotides within syntenic blocks; the remaining 2% positions could not be aligned unambiguously and were not

considered further. Almost two thirds of the 10,921 variations between the long-read assemblies overlapped with variations seen in the Illumina versus Nanopore-based assemblies and were preceded or followed by a homopolymer of three nucleotides or more (e.g. CCCCCC versus CCC.CC). These variations in homopolymer length likely result from erroneous base calling in the Nanopore data, as known to occur in the R9 flow cell series that was used in this study [61]. These variations were therefore not considered for subsequent analyses.

Repetitive sequences accounted for about 7% of the assemblies. In the CBS815.71-sp3 assembly, we identified 182 transposable element (TE) family consensus sequences: 4.2% Gypsy LTRs and 6.7% other LTRs, 7% non-LTR SINEs and LINEs, 7.5% TcMar-Fot1 DNA transposons, 3.4% other DNA transposons and the remaining 70.5% repetitive sequences could not be annotated (S2A Fig). Both genomes contained very few (<0.1%) signatures of RIP (repeat-induced point mutations), a fungal genome defense against transposable elements inducing C to T mutations in repeated sequences at target dinucleotides [62, 63].

## Structure of the mating-type locus in *Schizothecium tetrasporum*

The mating-type locus was located at about the 5.180 Mbp position on contig 1 of the CBS815.71-sp3 assembly, which likely corresponds to a fully assembled chromosome (Fig 4). The mating-type locus of *S. tetrasporum* had the typical structure of heterothallic *Sordariales* [20], with two highly divergent "alleles", referred to as idiomorphs, flanked by the *APN2* and *SLA2* genes (Figs 4C and S3, [64, 65]). The *MAT1-2* idiomorph, present in CBS815.71-sp3, harbored a single functional gene (*MAT1-2-1*), orthologous to the *mat+ FPR1* (*Pa_1_20590*) gene of *P. anserina* and the *mat_a-1* gene of *Neurospora crassa*. The other idiomorph, *MAT1-1*, present in CBS815.71-sp6, carried three genes (*MAT1-1-1*, *MAT1-1-2* and *MAT1-1-3*), orthologous and syntenic with the three genes present in the *mat-* idiomorph of *P. anserina* (*Pa_1_20593*, *_20592*, *_20591*) and the *mat_A* idiomorph of *N. crassa* (NCU01958, NCU01959, NCU01960). *MAT1-1-1* contained the typical MATα_HMG domain (pfam04769) and *MAT1-1-2* contained a HPG domain (pfam17043). *MAT1-1-3* and *MAT1-2-1* carried a MATA_HMG domain (cd01389). A sequence homologous to the 3' end of the *MAT1-1-1* coding sequence was also present next to the *MAT1-2* idiomorph. This truncated *MAT1-1-1* gene close to the *MAT1-2* idiomorph has a putative open reading frame starting 68 residues downstream of the start codon of the typical *MAT1-1-1* gene structure (S3 Fig). However, this putative start codon is in a poor context for translation initiation, with a pyrimidine in position -3 [64]. The truncated *MAT1-1-1* gene encodes most of the MATα_HMG domain and the second conserved domain adjacent to the MATα_HMG domain (S3 Fig; [66]). This gene was therefore named *TαD1* (truncated MATα_HMG domain). This name does not follow the standard mating-type gene nomenclature [57], as it is likely not part of the mating-type locus. We found very few RNAseq reads mapping on *TαD1* using the RNA-seq data generated for gene annotation (S4 Fig), supporting the view that *TαD1* is a pseudogene; its transcriptional status nevertheless remains to be investigated in other conditions. A similar truncated *MAT1-1-1* gene is present and has little or no expression in *MAT1-2* strains in the *Chaetomiaceae* (*Sordariales*) *Myceliophthora heterothallica*, *M. fergusii*, *M. hinnulea* and *Humicola hyalothermophila* [67]. It is unknown whether *TαD1* has any role in sexual reproduction.

## A region with stepwise recombination suppression around the mating-type locus in *Schizothecium tetrasporum*, located 1 Mb from the centromere

We plotted the synonymous divergence ($d_S$) between the two genomes of opposite mating types, for each gene along the contigs of the CBS815.71-sp3 assembly (Figs 5A and S5). Most of the $d_S$ values were zero, as expected in a selfing species with a high degree of homozygosity.

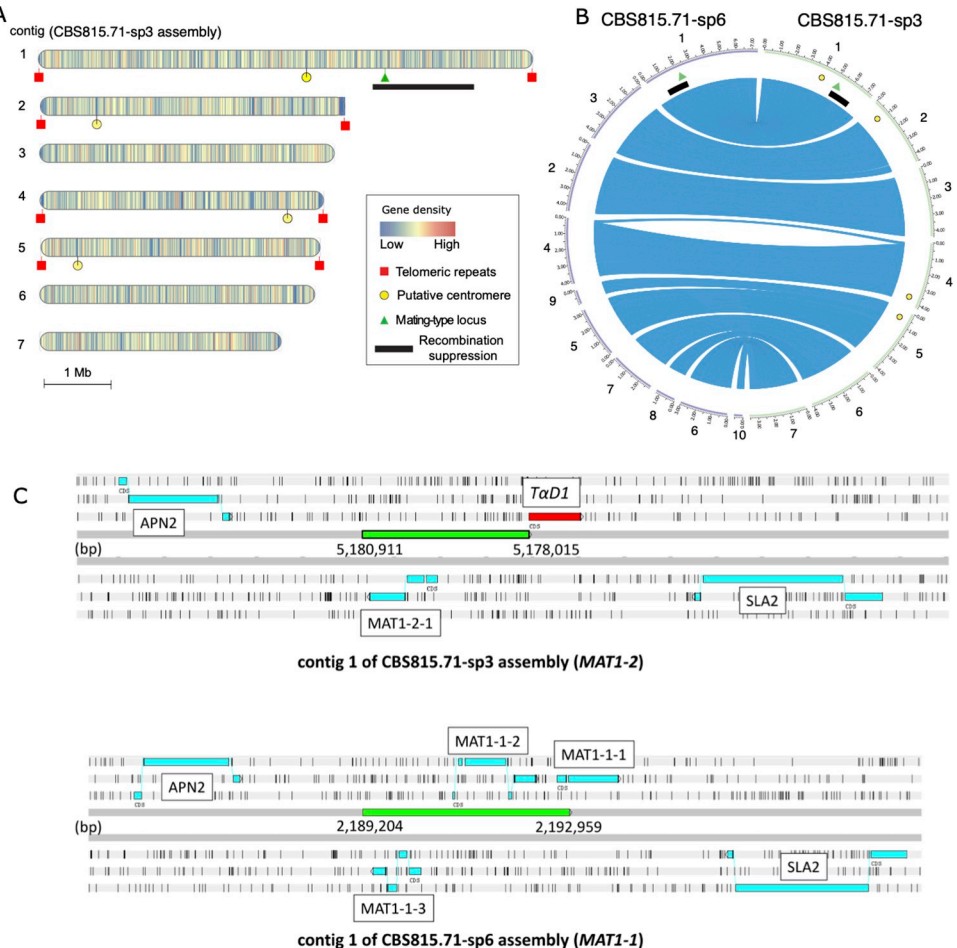

**Fig 4. Genome features and annotation of the mating-type locus in the *Schizothecium tetrasporum* CBS815.71 strain. A.** Ideograms representing the seven largest contigs of the CBS815.71-sp3 genome assembly. The mating-type locus is indicated with a green triangle, telomeric repeats with red boxes and putative centromeres with yellow circles. Gene density in 10 kb non-overlapping windows is represented by vertical bars on the contigs and the heatmap colors indicate low to high density. The non-recombining region around the mating-type locus is indicated by a black rectangle. **B.** Genome collinearity between opposite mating types of the *Schizothecium tetrasporum* CBS815.71 strain. Circular plot showing contigs from the CBS815.71-sp3 genome (green, right) and the CBS815.71-sp6 genome (purple, left). Only the largest contigs (>500 kb) were plotted. Blue links indicate the locations of orthologous genes (Mb). The mating-type locus is indicated with a green triangle. Putative centromeres and the non-recombining region around the mating-type locus identified on the CBS815.71-sp3 genome assembly are indicated by yellow circles and a black rectangle, respectively. **C.** Organization of the mating-type locus in CBS815.71-sp3 (top, *MAT1-2*) and CBS815.71-sp6 (bottom, *MAT1-1*) assemblies viewed in Artemis v18.0.2 [125]. The *APN2* and *SLA2* genes flanking the mating-type locus are indicated. The green boxes represent the nucleotide sequences differing between the two idiomorphs that were used to design markers for assessing mating type in the progeny. Coordinates of the region differing between *MAT1-1* and *MAT1-2* are indicated on contig 1 of the CBS815.71-sp3 assembly for *MAT1-2* and on contig 1 of the CBS815.71-sp6 assembly for *MAT1-1*. Contigs 1 of the CBS815.71-sp3 and CBS815.71-sp6 assemblies were assembled on opposite strands but for visualization purpose the same strand of both assemblies is shown on the figure. The *MAT1-1-1* pseudogene (*TαD1*) present in the CBS815.71-sp3 (*MAT1-2*) genome is boxed in red. The three forward and three reverse frames of the DNA sequence are presented; vertical bars indicate putative STOP codons.

Only around the mating-type locus did we find a cluster of genes with high d$_S$ values (Fig 5A), indicating recombination suppression. The high degree of homozygosity throughout the genome indicates that many selfing events occurred in the ancestors of the CBS815.71 hetero-karyotic strain; only the regions fully linked to the mating-type locus have remained heterozy-gous under selfing [31]. Indeed, for any gene linked to the mating-type locus, alleles associated

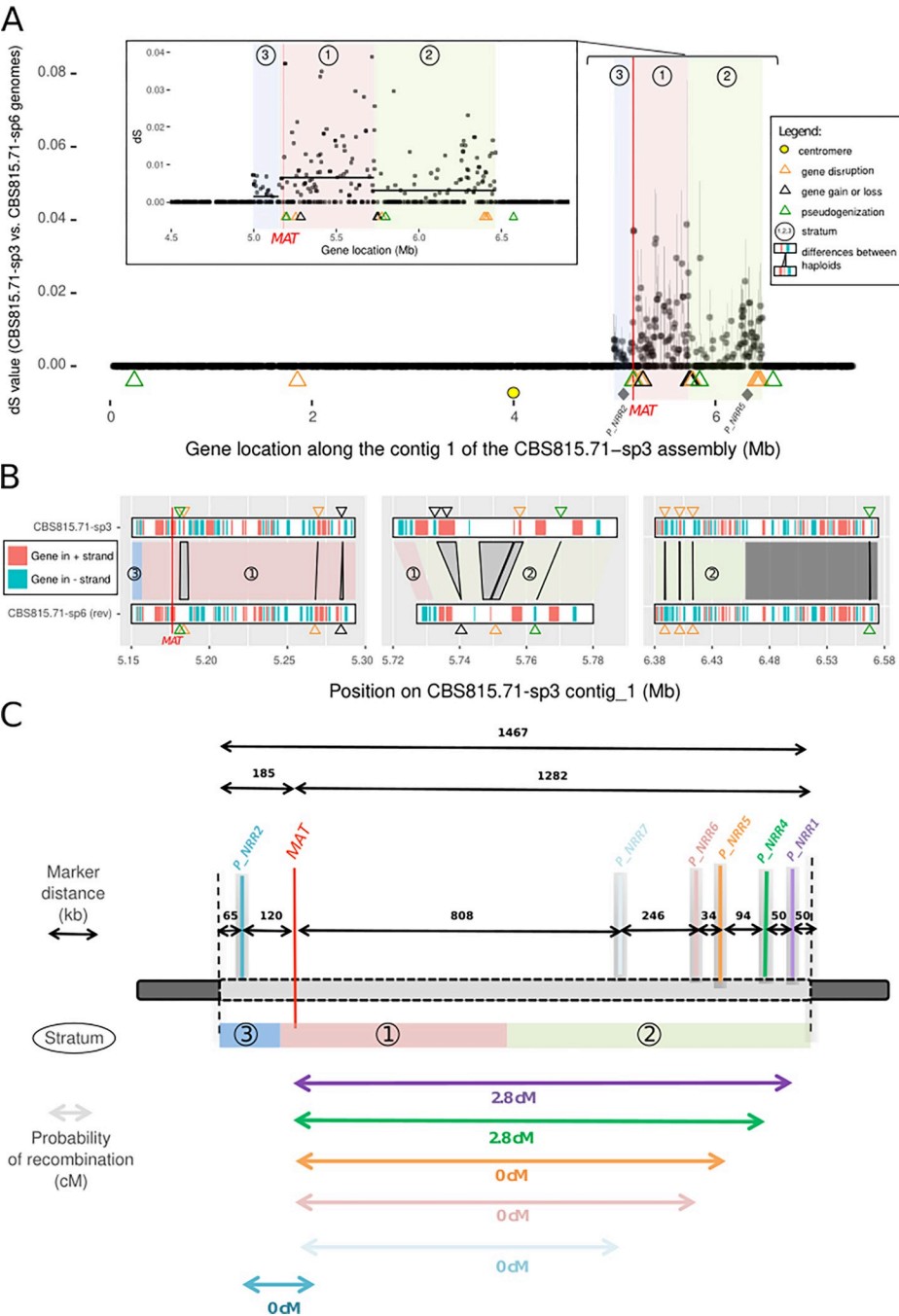

**Fig 5. Evidence for recombination suppression around the mating-type locus in the *Schizothecium tetrasporum* CBS815.71 strain. A.** Synonymous divergence (d_S) and its standard deviation along the mating-type chromosome in the *S. tetrasporum* CBS815.71 strain. d_S was calculated per gene, between the orthologs common to the CBS815.71-sp3 and CBS815.71-sp6 assemblies and was plotted, according to the CBS815.71-sp3 assembly gene position (contig 1), along the *x*-axis. The mating-type locus is indicated with a red vertical line. The putative centromere is indicated with a yellow circle. The *P_NRR2* and *P_NRR5* markers, which displayed no recombination event with the mating-type locus, are indicated with gray diamonds. The inset shows a zoom on the three strata, with d_S values in the region located between 4.5 Mb and 6.5 Mb of the contig 1; black horizontal bars represent the mean d_S values of evolutionary strata 1, 2, and 3, respectively. In both plots, evolutionary strata 1, 2, and 3 are indicated by red, green and blue rectangles, respectively, and their numbers in circles. Gene disruption, gene gain or loss and pseudogenization events are indicated by orange, black and green triangles, respectively. **B.** Zoom in the regions with putative deleterious mutations identified as synteny breaks near strata edges. Links between CBS815.71-sp3 and CBS815.71-sp6 assemblies correspond to gene-disrupting, pseudogenizing or gain/loss events induced by indels, point mutations disrupting the

stop codon or extreme sequence divergence. The original figure obtained with pafr is shown in S9 Fig. Symbol and color shading key are the same as in panel A, with differences between haploid genomes shown by black vetical lines and grey rectangles. C. Probability of recombination in centimorgans (cM) between markers and the mating-type locus in the heterokaryotic strain CBS815.71 after one round of experimental selfing measured in 107 offspring. The non-recombining region and the evolutionary strata 1, 2, 3 defined based on $d_S$ values in the heterokaryotic strain CBS815.71 are shown with red, green and blue rectangles, respectively and their numbers in circles. Relative locations of the mating-type locus (MAT) and markers tested (*P_NRR1*, *P_NRR2*, *P_NRR4*, *P_NRR5*, *P_NRR6*, *P_NRR7*) are shown with vertical bars. Genomic distances (in kilobases) between markers are indicated above black arrows.

with the alternative mating types gradually accumulate different mutations over the time since recombination suppression. Synonymous divergence ($d_S$) can, thus, be used as a proxy for the age of the linkage between a gene and the mating-type locus [7].

The cluster of genes with high $d_S$ values around the mating-type locus spanned a 1.47 Mb region located between 4.99 Mbp and 6.46 Mbp on scaffold 1 of the CBS815.71-sp3 assembly, encompassing 404 predicted genes in the CBS815.71-sp3 assembly and 411 predicted genes in the CBS815.71-sp6 assembly. We found no predicted function potentially directly related to mating compatibility for any of the 404 genes in the non-recombining region of the CBS815.71-sp3 assembly other than the mating-type locus. A total of 15 genes in the non-recombining region encode proteins that were identified as putative transcription factors or involved in signaling pathways (S3 Table). The four orthologs of the *P. anserina* genes located in its non-recombining region and known to be involved in post-mating sexual development (but not mating compatibility) were also found in the non-recombining region in *S. tetrasporum*: *ami1*, controlling nuclear migration events [68], *PaRID*, a DNA methyltransferase controlling sexual development in *P. anserina* [69], *AS4*, encoding the eEF1A translation elongation factor, and *su12*, encoding a ribosomal protein, mutations of which often result in sterility in *P. anserina* [70, 71].

The $d_S$ values plotted along the mating-type chromosome seemed to indicate the existence of three distinct evolutionary strata, *i.e.* adjacent regions with different differentiation levels. We indeed detected a central region of high $d_S$ values (hereafter called stratum 1), flanked by two regions with lower but non-zero $d_S$ values (called strata 2 and 3). The mating-type locus was located close to the limit between stratum 1 (the oldest stratum) and stratum 3 (Figs 5A and S6). We tested the existence of three evolutionary strata more formally, by splitting the heterozygous region with sliding limits in the CBS815.71-sp3 assembly and calculating, for each limit, the difference in $d_S$ values between the two segments; limits between strata should correspond to a peak in $d_S$ value differences according to stratum definition [31]. We first arbitrarily set the limit of the first stratum by eye at 5,180,000 bp, i.e. at the mating-type locus location, and applied the automatic procedure to the rest of the heterozygous region. The $d_S$ values peaked at 5,730,660 bp, identifying the limit between strata 1 and 2 (S6 Fig). We then re-applied the procedure to the part of the heterozygous region without stratum 2. The $d_S$ values peaked at 5,153,200 bp, identifying the limit between strata 1 and 3 at 30 kb from the mating-type locus (S6 Fig). The median $d_S$ values were significantly different between strata 1 and 2 (pairwise Wilcoxon test *p*-value = 0.00052 with the false discovery rate (FDR) correction) and between strata 1 and 3 (pairwise Wilcoxon test *p*-value = 0.00148 with FDR correction), supporting the existence of distinct evolutionary strata. Median $d_S$ values did not differ significantly between strata 2 and 3, located on opposite sides of stratum 1 (pairwise multiple-comparison Wilcoxon test *p*-value = 0.49604 with FDR correction). Stratum 1 spanned 577 kb, stratum 2 spanned 732 kb and stratum 3 spanned 158 kb (S6 Fig). Stratum 1 (the oldest stratum) encompassed the mating-type locus.

We looked for the centromere on the mating-type chromosome because its position may provide important insight into the segregation of the mating-type locus at the first meiotic

division. We found a single location with a drop in GC content along the mating-type chromosome of the CBS815.71-sp3 assembly outside of the telomeric regions, with 53.5% GC between 3.990 Mbp and 4.025 Mbp, versus the genome-wide mean value of 58.76% (S7 Fig). A drop in GC content can pinpoint the centromere in some fungi, such as some *Neurospora* spp. in which GC content drops to about 44%, whereas the genome-wide mean value is about 53% [72]. The lower variation of GC content observed in the CBS815.71-sp3 genome assembly may be due to a lack of RIP in *S. tetrasporum*. For the other three fully assembled chromosomes of the CBS815.71-sp3 genome, similar drops in GC content were also identified at a single location per chromosome, co-localizing with clusters of repeats. We characterized the repeat content of these four putative centromere regions. We found no specific repeats, but we did detect an enrichment in *TcMar-Fot1* DNA transposons relative to the genome-wide mean (S2A Fig). The putative centromere of the mating-type chromosome was located about 1 Mb from the edge of the non-recombining region around the mating-type locus (Fig 5A).

We found no enrichment in repetitive sequences in the non-recombining region around the mating-type locus relative to autosomes and the recombining regions on the mating-type chromosome in the CBS815.71-sp3 assembly. The mean repetitive sequence densities calculated in 10 kb windows overlapping by 1 kb were 2.52%, 5.05% and 4.21% in the non-recombining region around the mating-type locus, the autosomes and the recombining regions on the mating-type chromosome, respectively; pairwise Wilcoxon tests revealed no significant differences in repetitive sequence densities. We found no difference in TE family proportions between these regions either (S2B Fig). Repetitive sequence density was even lower in strata 1 and 3 compared to autosomes and stratum 2 (*e.g.* stratum 3 vs 2: *p*-value for a pairwise Wilcoxon test = 0.0179 with FDR correction; stratum 1 vs 2 *p*-value for a pairwise Wilcoxon test = 0.0025 with FDR correction). We found no genes with a $d_N/d_S$ ratio $>1$ in the non-recombining region, *i.e.* with higher non-synonymous than synonymous substitution rates (the mean $d_N/d_S$ was 0.12 for genes with non-zero $d_S$ values). As noted above, we found no evidence for genomic rearrangements between the two mating-type chromosomes in the non-recombining region (Fig 4B).

We however found signs of degeneration in terms of gene losses and gene disruptions in the non-recombining region: we detected in total 26 differences between the two genomes of opposite mating types in terms of presence/absence of predicted genes (*i.e.* synteny breaks), six of which occurred in the 1.47 Mb of the non-recombining region, which was significantly more than expected under random distribution (Fisher's exact test p-value = 0.0003). Global alignment of MAT-containing contigs revealed that the non-recombining region encompassed three gene gains/losses due to large indels, two unalignable regions in the non-recombining region, including the mating-type idiomorphs, four gene disruptions associated with small indels in the coding regions and one pseudogenization due to a mutation in a stop codon (Fig 5A and 5B). All but one of the remaining 20 gene presence/absence differences between the two genomes of opposite mating types were associated with non-sense mutations or gene-disrupting frameshifts introduced by short indels. Interestingly, the gene gains/losses and gene disruptions were specifically located at the strata edges (Fig 5A and 5B), pointing more to a cause than a consequence of recombination suppression, as deleterious mutations following recombination suppression should occur all along the non-recombining region. A pseudogene was also found in the recombining region flanking the non-recombining region (Fig 5A and 5B).

## Validation of the lack of recombination by progeny analysis

We investigated the lack of recombination in the heterozygous region around the mating-type locus in the CBS815.71 strain. We obtained offspring resulting from a selfing event in the

CBS815.71 strain, *i.e.* a cross between CBS815.71-sp3 and CBS815.71-sp6. We genotyped 107 offspring resulting from haploid spores isolated from five-spore asci (*i.e.* asci with three large dikaryotic spores and two small monokaryotic spores; Fig 2). We initially used two genetic markers, one at each end of the heterozygous region (Fig 5C). We detected no recombination events at the edge of stratum 3 (between the *P_NRR2* marker and the mating-type locus) but we found recombination events at the edge of stratum 2 (between the markers *P_NRR1*, *P_NRR4* and the mating type locus) in three of the 107 offspring. We therefore used three additional markers, in the three offspring in which we detected recombination with the initial markers, to delimit more finely the regions in which recombination events occurred (Fig 5C). We found no recombination event between the mating-type locus and the additional markers, *P_NRR7, P_NRR6* and *P_NRR5*, located 0.808 Mb, 1.054 Mb and 1.088 Mb from the mating-type locus, respectively (Fig 5C). Given the position of the mating-type locus, this suggests a complete lack of recombination over a 1.37 Mb region in strata 1 and 3, and in a large part of stratum 2, as recombination occurred only at the edge of this stratum. Stratum 2 is, therefore, likely slightly smaller than the heterozygous region in the CBS815.71 strain, the heterozygosity at its edge probably resulting from an outcrossing event a few generations ago.

## Orthologs in the non-recombining regions across Sordariales

An alternative to the deleterious mutation-sheltering hypothesis to explain evolutionary strata is the antagonistic selection or supergene hypothesis, which postulates that recombination suppression is selected for maintaining beneficial allelic combinations [73]. This hypothesis could be supported if the same genes were located in the non-recombining region in the various pseudo-homothallic lineages despite extensive rearrangements in other genomic regions, as this may indicate a particular advantage to link these genes to the mating-type locus. We therefore investigated gene orthology relationships and synteny between the non-recombining regions on the mating-type chromosomes of the three pseudo-homothallic species *S. tetrasporum*, *P. anserina* and *N. tetrasperma*. However, several rearrangements occurred following recombination suppression on the mating-type chromosomes in *N. tetrasperma*. We therefore used the genome of the closely related heterothallic *N. crassa* species instead, as a proxy for the ancestral gene order in *N. tetrasperma* before recombination suppression. No recombination suppression around the mating-type locus has been reported in *N. crassa* [30, 37]. This analysis therefore also provided an opportunity for an interesting contrast with species not displaying recombination suppression.

In *S. tetrasporum*, 261 of the 404 genes located in the identified region of suppressed recombination around the mating-type locus had orthologous genes in both *P. anserina* and *N. crassa*. We found that 138 of these 261 genes were also located in the region of suppressed recombination around the mating-type locus in the *P. anserina* S strain. Most of these shared genes (74%) were part of stratum 1, the oldest evolutionary stratum in *S. tetrasporum*, encompassing the mating-type locus (Fig 6A and 6B (ii)). No evolutionary strata have previously been identified in *P. anserina*, but the oldest evolutionary stratum in *P. pseudocomata*, a cryptic species closely related to *P. anserina* [31], corresponded mainly to the stratum 3 and a part of the stratum 1 in *S. tetrasporum*. Most genes (82%) of stratum 2 in *S. tetrasporum* had orthologs in *P. anserina* in the recombining region between the mating-type locus and the centromere (Fig 6B(ii)). The region of recombination suppression was located, when starting from the mating-type locus, in the direction of the telomere in *S. tetrasporum*, whereas it was located, when starting from the mating-type locus, in the direction of the centromere in *P. anserina* and other species of the *P. anserina* species complex (Fig 6B(ii)). The distance between the centromere and the mating-type locus was 1.1 Mb in *S. tetrasporum* and 2.9 Mb in *P. anserina*.

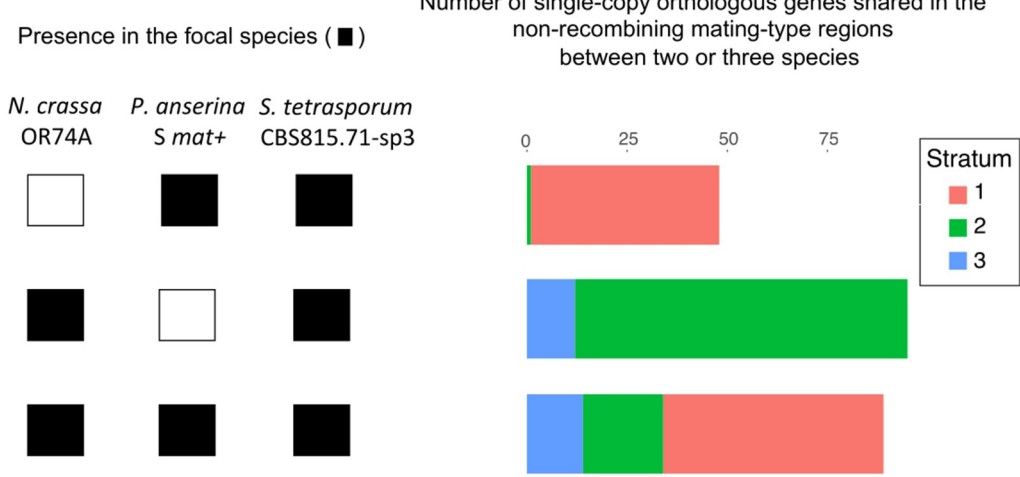

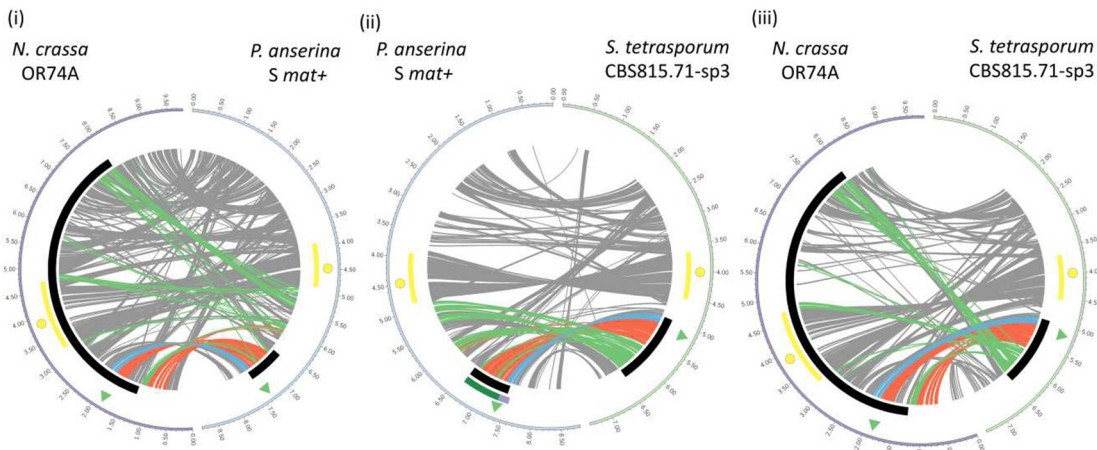

**Fig 6. Shared gene content and collinearity between the mating-type chromosomes of *Schizothecium tetrasporum*, *Podospora anserina* and *Neurospora tetrasperma*. A**. Number of single-copy orthologous genes present in the non-recombining region of *S. tetrasporum* that are also in the non-recombining regions of *P. anserina* and/or *N. tetrasperma*, as indicated by the squares at the bottom (a filled square indicates the presence in the non-recombining region of the focal species). **B.** Collinearity between (i) *N. crassa* (left) and *P. anserina* (right); (ii) *P. anserina* (left) and *S. tetrasporum* (right); (iii) *N. crassa* (left) and *S. tetrasporum* (right). Collinearity is based on single-copy orthologous genes. Centromere is indicated with a yellow circle and the region of well-conserved gene order around the centromere is highlighted with a yellow rectangle. The mating-type locus location is shown with a green triangle and the non-recombining region around the mating-type locus in *P. anserina* [27] and *S. tetrasporum* with a black track. Along the recombining mating-type chromosome of *N. crassa* a black rectangle indicates the orthologous region that is non recombining in multiple *N. tetrasperma* lineages [30]. Red, green and blue links highlight orthologous genes located in the evolutionary strata 1, 2, and 3 of *S. tetrasporum*, respectively. Grey links indicate other orthologous genes. On panel (ii), the evolutionary strata identified previously in the *P. pseudocomata* CBS415.72 strain, that is closely related to the *P. anserina* S strain, are indicated by purple (for the oldest stratum) and dark green (for the youngest stratum) tracks [31], at left.

We further found that 185 genes in the non-recombining region of *S. tetrasporum* were also present in the non-recombining region in *N. tetrasperma*, with the genes from *S. tetrasporum* strata 1 and 2 present in the flanking region of the non-recombining region in *N. tetrasperma* (Fig 6B(iii)). In total, 90 genes were common to the non-recombining regions of the three pseudo-homothallic species (Fig 6A), mainly genes from a part of *S. tetrasporum* stratum 1 (Fig 6B). The predicted functions of these 90 orthologs, when available, did not appear to be related to mating-type functions or to be able to provide differential fitness for the two mating types, *i.e.* to be under antagonistic selection according to mating type (S4 Table).

By visual inspection of single-copy orthologous gene locations between species, we found that the mating-type chromosomes, as well as all the other chromosomes, displayed regions of synteny, but also breaks of synteny between species or mesosynteny, *i.e.* a global conservation of gene content despite rearrangements in order and orientation (Figs 6B and S8), as expected given their phylogenetic distance and as previously shown between *P. anserina* and *N. crassa* [74].

Overall, we thus found some degree of shared gene content in the non-recombining regions of the three distant pseudo-homothallic lineages, but we found no function that could be under antagonistic selection in the alternative mating types and no exceptional synteny or shared content around the mating-type locus compared to the genomic background, which therefore brings no support for the supergene hypothesis. Furthermore, the synteny was also relatively conserved in this region in other non-pseudo-homothallic *Sordariales*, as shown by the comparison with *N. crassa* (Fig 6B) and as previously reported [74]. The region around the putative centromere identified on the *S. tetrasporum* mating-type chromosome had orthologous genes and gene order conserved with the *P. anserina* and *N. crassa* centromeric region over a distance of 700 kb despite an inversion, providing support for the putative location of the centromere on the *S. tetrasporum* mating-type chromosome identified above (Fig 6A).

## Discussion

### Evolutionary convergence of a pseudo-homothallic breeding system and recombination suppression across distant fungal lineages

This study of the breeding system and sexual biology of the little-studied pseudo-homothallic fungus *S. tetrasporum* reveals a striking convergence with *P. anserina* and other cryptic species of the *P. anserina* complex, for multiple ecological and reproductive traits. These two distantly related fungi thrive in herbivore dung and produce predominantly dikaryotic spores carrying opposite mating types. In both cases, this is due to mating-type segregation during the second meiotic division associated with a single and systematic CO event between the centromere and the mating-type locus. In *S. tetrasporum*, this was shown by cytological observations in a previous study [25], and confirmed here by the existence of a large region with zero $d_S$ values between the centromere and the mating-type locus, indicating recombination between the centromere and the mating-type locus. This situation is similar as in *P. anserina* [26, 30, 31]. In addition, both *S. tetrasporum* and *P. anserina* display recombination suppression around this locus ([27] and this study). Further studies on additional strains of *S. tetrasporum* will be interesting to assess whether the region of suppressed recombination is always present and always of the same size. We found that recombination suppression had expanded in several successive steps in the studied strains of *S. tetrasporum*, as in *P. pseudocomata*, which belongs to the *P. anserina* species complex [31]. The association between pseudo-homothallism (self-fertile dikaryotic spores), recombination suppression around the mating-type locus and evolutionary strata also evolved independently in the *N. tetrasperma* complex, representing an additional case of convergence. However, unlike *S. tetrasporum* and *P. anserina*, *N. tetrasperma*

displays recombination suppression up to the centromere and mating-type segregation during the first meiotic division (Fig 1; [30, 72]).

## Association between pseudo-homothallism and recombination suppression: evolutionary and proximal causes

Such a convergent association between recombination suppression and pseudo-homothallism suggests causal relationships. A first hypothesis is that the suppression of recombination would be linked to the still unknown mechanism that imposes exactly one CO between the centromere and the mating-type locus. The selective advantage of producing self-fertile spores would then be the driver of recombination suppression, allowing reproductive insurance in short-lived habitats such as dung. However, the occurrence of a CO between the centromere and the proximal end of the region of suppressed recombination in almost all meiosis I to cause mating-type segregation during the second meiotic division still calls for an explanation. Indeed, if we consider the 21 ± 3 COs occurring at each meiosis in one nucleus of *Sordaria macrospora* [75, 76] as a proxy for the number of COs occurring in *S. tetrasporum*, displaying approximately the same genome size, and if we take into account the number of obligate COs occurring on each of the seven chromosomes and ensuring proper segregation, we can estimate that 11 to 17 COs remain to be randomly distributed across a genome of 32 Mbp. It is therefore unlikely that one CO takes place between the centromere and the mating-type locus in 99.5% of the meioses in *P. anserina*, the distance between the centromere and the mating-type locus being 2.9 Mbp, and in *S. tetrasporum*, the distance between the centromere and the mating-type locus being 1.2 Mbp. The occurrence of a second CO between the centromere and the mating-type locus has to be prevented, as two COs in this region would result in FDS in half of the cases. This second CO might be prevented by positive interference, but this mechanism alone may not be sufficient to reach the very high frequency of SDS. It therefore appears more likely that a yet unidentified mechanism has evolved that enforces the occurrence of the single and systematic CO event between the centromere and the mating-type locus, independently in both *P. anserina* and *S. tetrasporum*, together with a mechanism compacting two nuclei per ascospore, for the production of self-fertile ascospores. This type of CO is different from the concept of "obligate COs", which refers to the fact that, in most species, each chromosome undergoes at least one CO [77, 78]. Such "obligate COs" do not occur at a specific location, except between the X and Y chromosomes in many mammals [79].

A second hypothesis to explain the evolution of recombination suppression can be proposed considering that the mating-type locus is always heterozygous in heterokaryons. Indeed, recombination suppression capturing a permanently heterozygous and bi-allelic locus may be selected due to its capacity to shelter recessive deleterious alleles in less-loaded genomic regions, in organisms with an extended diploid or dikaryotic phase [16]. More precisely, recombination suppression may be selected because it protects DNA fragments with fewer deleterious mutations than average in the population, but the load of these mutations will still prove problematic when they become frequent enough to become homozygous. The capture of a permanently heterozygous allele prevents homozygosity, allowing fixation of recombination suppression [16]. This mechanism can also explain the evolution of successive evolutionary strata, as observed here in *S. tetrasporum*, and previously in *N. tetrasperma* and *P. pseudocomata* [26, 31, 72], but also in the cultivated variety of the basidiomycete mushroom *Agaricus bisporus* var. *bisporus* [50], and repeatedly in *Microbotryum* basidiomycete fungi, plant pathogens in natural ecosystems [8, 9, 18, 80]. All these lineages undergo mostly or only automixis (intra-tetrad mating) and have only two mating types, which has been theoretically shown to favor the sheltering of deleterious alleles near the mating-type locus [16, 47, 48].

In this study, we found signs of gene losses and disruptions in the non-recombining region around the mating-type locus. The gene losses and disruptions were found precisely at the edges of evolutionary strata, which supports the deleterious mutation hypothesis: if deleterious mutations are only a consequence of recombination suppression, they are expected to occur all along the non-recombining region. Their locations at edges of strata suggest that recombination suppression evolved to shelter the deleterious mutations. We also found a gene loss just at the margin of the non-recombining region, as also predicted by the model showing that a mating-type locus can protect nearby deleterious mutations, especially under automixis [48]. We found no detectable differences in growth between homokaryons of the two mating types and the heterokaryon, but fitness differences even smaller than those detectable here can promote the evolution of recombination suppression [16]. In the deleterious allele sheltering hypothesis, the non-recombining fragment carries in the first stage very few deleterious alleles, fewer than average in the population in the corresponding region [16], so that their fitness effect may be difficult to detect experimentally. Furthermore, the fitness advantage of the non-recombinant fragment is expected as a mean across all individuals in the population [16], and is therefore not necessarily expected when studying, as here, a single strain. In addition, genes with deleterious alleles may be involved in specific parts of the life cycle not evaluated *in vitro*, *e.g.* related to the dung habitat or resistance to harsh conditions. We found that 13% of the thalli grown from ascospores *in vitro* ended up homokaryotic, but the mechanism involved could function in haplodiplontic lifecycles, depending on the mean length or frequency of the haploid phase and the strength of selection [16]. The mycelia in dung may not grow for long enough for the loss of a nucleus to occur very often, especially as *S. tetrasporum* is mostly found on small pellets of rabbit dung [81].

A third hypothesis to explain recombination suppression near the mating-type locus could be a neutral one, as proposed earlier [8, 9], in which inversions or other recombination modifiers would be neutral in an initial stage at the margin of an already non-recombining region (the mating-type locus itself) and could fix by drift. A fourth hypothesis would be a selection of recombination suppression to maintain beneficial allelic combinations [73], as observed for supergenes in plants and animals [82, 83]. Despite the finding of a number of shared orthologous genes in the non-recombining regions in *P. anserina*, *N. tetrasperma* and *S. tetrasporum*, our data did not bring support for the supergene hypothesis, as the overall synteny was not higher than in the rest of the genome and the gene functions were not related to mating-type antagonistic functions.

These various evolutionary hypotheses for explaining recombination suppression are not mutually exclusive, and the evolutionary causes of recombination suppression may differ between evolutionary strata and species. The proximal (mechanistic) causes of recombination suppression also remain unknown. We detected no chromosomal inversion between mating-type chromosomes in *S. tetrasporum*, as in some other fungi in which recombination is also suppressed close to the mating-type locus without inversion, such as *P. anserina* and other species of the *P. anserina* complex, *Microbotryum* fungi and some strains of *N. tetrasperma* [8, 27, 30, 31]. Histone modification or methylation can also prevent recombination [84, 85] and this aspect warrants investigation in *S. tetrasporum*. There may also be recombination modifiers in *cis*. The molecular cause for the single CO event between the centromere and the mating-type locus is also unknown, in both *S. tetrasporum* and *P. anserina* [27]. The shorter genomic region between the centromere and the mating-type locus in *S. tetrasporum* (1.2 Mb) than in *P. anserina* (2.9 Mb) makes this species an excellent model for identifying the proximal factors responsible for the single CO event, if they are located in *cis*. The occurrence of a single systematic CO event between the mating-type locus and the centromere in these two species provides an opportunity to study the genetic basis of the convergent evolution of this phenomenon. In

*P. anserina*, the systematic CO does not occur at a precise or localized hotspot [31, 86, 87] and it will be interesting to assess in future studies whether this is also the case in *S. tetrasporum*.

## Evolutionary strata

We detected three evolutionary strata in the CBS815.71 *S. tetrasporum* strain, indicating a stepwise extension of recombination suppression, which has been even more puzzling in evolutionary biology than a single step of recombination suppression [4, 9–11]. Our progeny analysis showed that strata 1 and 3, and much of stratum 2, probably display a full suppression of recombination. This gradual progression of recombination suppression cannot be explained by sexual antagonism, as male and female functions are not associated with mating types in fungi [19, 88, 89]. In addition, monokaryotic ascospores are rare in *S. tetrasporum* and traits and ecological features are similar between mating types, so other types of antagonistic selection are also unlikely. Stepwise recombination suppression in this dikaryotic fungus may have been selected to shelter deleterious alleles as discussed above [16]. Actually, we did identify gene losses in the non-recombining region, suggesting the existence of deleterious effects at some stages in the life cycle. These deleterious mutations were strikingly located at the borders of the different evolutionary strata, suggesting that they are a cause rather than a consequence of recombination suppression. A pseudogenization event was also found within the recombining region but near the non-recombining region, as predicted by a recent theoretical model showing that a mating-type locus can shelter deleterious mutations at its margin in automictic species, allowing their accumulation there, which then can further foster recombination suppression expansion [16, 47, 48]. This gene might be already trapped in the non-recombining region in other strains and this prediction could be checked in future studies by sequencing further genomes.

Alternatively, the different strata may correspond to gradual increases in the SDS frequency, by stepwise increase of the efficiency of the mechanism preventing a second CO in the region of mating-type locus and FDS. The selective pressure for this stepwise increase would be to reach a higher number of dikaryotic ascospores growing into self-fertile mycelia. The evolutionary strata could also be caused by the neutral fixation of recombination modifiers by drift near an already non-recombining region [8, 9].

## Signs of degeneration in the non-recombining region

We found signs of degeneration in the non-recombining region in terms of gene losses and gene disruptions. In contrast, we found no enrichment in repeats, rearrangements or non-synonymous substitutions, whereas transposable elements, non-synonymous substitutions and non-optimal codon usage have all been found to increase with time since recombination suppression in the mating-type chromosomes of *Microbotryum* anther-smut fungi [9, 46, 80]. The suppression of recombination may have occurred too recently in *S. tetrasporum* for these marks of degeneration to evolve or there may be an efficient mechanism of TE control. We found no footprints of RIP, the fungal genome defense mechanism that introduces cytosine-to-thymine transitions into repetitive sequences to control transposable elements [62, 63]. Nevertheless, the TE load was not high in *S. tetrasporum*, suggesting that this fungus may have evolved another mechanism of transposable element control, such as RNA silencing [90]. Lack of transposable element accumulation has also been reported in the non-recombining region of *P. anserina*, with also a lack of gene losses [27]. Gene conversion has been suggested to account, at least partly for this lack of degeneration, which may also apply to *S. tetrasporum* [27].

## Mating system in *Schizothecium tetrasporum*

We found a higher percentage of homokaryotic thalli resulting from ascospore germination in *S. tetrasporum* than in *P. anserina* (13% versus 1%). Homokaryons may be recovered due to first-division segregation, the loss of one nucleus after germination or a particular pattern of nucleus migration following the post-meiosis mitosis. We found that nuclei could be lost during vegetative growth *in vitro* in *S. tetrasporum* (here in 13% of the germinating ascospores), whereas this is not the case in *P. anserina* [27], suggesting that *S. tetrasporum* may sometimes, albeit rarely, end up homokaryotic in nature, and may thus be facultatively pseudo-homothallic/heterothallic. Such nucleus losses may be due to active competition between nuclei [91, 92] or to the presence of the identified gene losses. The occurrence of mycelia of a single mating type could promote some degree of outcrossing, which could be checked in analyses of the levels of heterozygosity of dikaryons in natural populations. Nevertheless, the rate of self-fertile (heterokaryotic) ascospores was high, and this would be expected to promote a high level of selfing. Population genomics approaches detected widespread genome-wide homozygosity in dikaryons of *P. anserina*, indicating high selfing rates, but with occasional stretches of heterozygosity, also indicating the occurrence of outcrossing in nature [31, 93, 94]. The optimal media and conditions for vegetative growth and sexual reproduction identified here will facilitate further investigations of the lifecycle and biology of *S. tetrasporum*. The reference strain was found to be highly homozygous, as expected under high rates of selfing. If we assume that the non-recombining region evolved to prevent a second CO, in order to maximize the frequency of SDS and thereby of self-fertile spores, it may appear intriguing that no mechanism evolved to subsequently maintain a dikaryotic mycelium, as for instance the crozier cells that are present in the perithecia and are analogous to the dikaryotic clamp cells of Basidiomycetes [95]. It may be that, in nature, the mycelium does not grow long enough to frequently loose a nucleus or that the occurrence of mycelia carrying a single mating type is beneficial in promoting outcrossing, as does the formation of a few asci with five ascospores including two monokaryotic ascospores.

## Conclusion

In conclusion, we found a striking convergence in lifecycle, sexual biology and stepwise recombination suppression around the mating-type locus in phylogenetically distant fungi of the order *Sordariales*. Documenting convergent evolution is important, as it shows that evolution can be repeatable, and not only contingent [96]. The broad convergence towards an association between dikaryotic life cycle and stepwise recombination suppression beyond the mating-type locus despite the lack of sexual antagonism provides support for alternative hypotheses, such as deleterious allele-sheltering, as explanations for the suppression of recombination beyond mating-type or sex-determining alleles [16]. The data obtained here provide the first experimental indication of a causal link between the sheltering of deleterious alleles and the stepwise extension of non-recombining regions. However, it does not provide definitive evidence for this or another hypothesis. Studying recombination suppression in additional heterothallic and pseudo-homothallic fungal lineages in the future will allow further testing the strength of the association between evolutionary strata and pseudo-homothallism, which is a prediction of the deleterious mutation-sheltering hypothesis. Studying recombination suppression in heterothallic fungi with bi-allelic versus multi-allelic mating-type loci and with contrasting lengths of their dikaryotic phase could further allow testing the deleterious allele-sheltering hypothesis, with the prediction that evolutionary strata should be more frequent with more extended dikaryotic phases and bi-allelic mating-type loci.

## Materials and methods

### Strain origins, experimental crosses and culture

The dikaryotic strain CBS815.71 of *S. tetrasporum* was obtained from the Centraalbureau voor Schimmelcultures (CBS). No optimized methods have been described for culturing *S. tetrasporum* and inducing sexual reproduction in this species. We therefore tested several media, to select those optimal for growth and sexual reproduction (M2, M3, Oat Meal Agar and V8 (France) agar; [24]), as well as for ascospore germination (the same media plus *P. anserina* germination medium (G), [24]). Growth and sexual reproduction were optimal on V8 medium (composition per liter: 15 g agar, 2 g CaCO$_3$, 200 mL *V8* tomato juice and 800 mL water) at 22°C with 12 h of illumination. Incubation at a higher temperature, such as 26°C, often resulted in aborted fruiting bodies. Growth and ascospore recovery were very slow at lower temperatures (*e.g.*, 18°C). We observed 100% germination on G medium (composition per liter: 4.4 g ammonium acetate, 15 g bactopeptone and 13 g agar) with a 65°C heat shock for 30 minutes just after the plating of ascospores. On all the other media tested, germination rates were lower, with no germination in the absence of a heat shock. In optimal conditions, mature perithecia ejecting ascospores were obtained three weeks after inoculation on V8 medium, and germination occurred overnight on G medium after a heat shock. Ascospores could be collected and plated with the same methods as for *P. anserina* [24].

In these optimal conditions, we generated and isolated two F1 monokaryotic strains by inducing a sexual cycle in the self-fertile heterokaryotic culture received from CBS815.71: the CBS815.71-sp3 strain with a mating-type allele orthologous to the *mat+* mating type of *Podospora anserina*, corresponding to the *MAT1-2* allele according to the official nomenclature [57], and the CBS815.71-sp6 strain with a mating-type allele orthologous to the *mat-* mating type of *P. anserina*, corresponding to the *MAT1-1* allele according to the official nomenclature [57]. We assessed the frequency of second-division segregation (SDS) at the mating-type locus and, therefore, the frequency of heterokaryotic spore formation, by generating and isolating 190 four-spore asci from a cross between CBS815.71-sp3 and CBS815.71-sp6. We studied the size of the region in which recombination was fully abolished around the mating-type locus, by isolating 107 small ascospores from five-spore asci (monokaryotic) from a cross between CBS815.71-sp3 and CBS815.71-sp6. After isolation, the strains were maintained on M2 minimal medium to prevent contamination [24], and were incubated at 22°C for growth. The growth of CBS815.71-sp3 and CBS815.71-sp6 homokaryons and the CBS815.71 dikaryotic strain was assessed on OA, V8 and M2 media [24]. The diameter of mycelial colonies was measured after 3 and 10 days of growth at 22°C with 12 h light, with a 10 cm ruler. Mean growth rate per day was calculated over the seven-day period from day 3 to day 10. Dry weights were measured after 10 days of growth in liquid M2 medium (50 mL); the mycelium was filtered and dried at 65°C overnight. Measurements were performed on five Petri dishes for each strain and each medium. ANOVA was performed with R software v 4.2.0.

### Genome sequencing and other genomic data

We generated long read-based genome assemblies by sequencing the genomes of the *S. tetrasporum* strains CBS815.71-sp3 and CBS815.71-sp6 with both Illumina technology on the HiSeq2500 platform (150 bp long paired-end reads) and Oxford Nanopore MinION technology with an R9 flow cell. For each DNA extraction, we prepared 100–200 mg aliquots of mycelium, which were stored in a 1.5 mL Eppendorf tube at -80°C for at least 24 hours before lyophilization. A sheet of cellophane was placed over the M2 medium before inoculation with spores, such that the mycelium grew on the cellophane for three to four days and could be

scraped off easily. For Illumina sequencing, DNA was extracted with the commercial Nucleospin Soil kit from Macherey Nagel. Illumina library preparation and genome sequencing were outsourced to the sequencing core facility of Genoscope (last accessed May 15, 2021). For Nanopore sequencing, mycelium DNA was extracted with the NucleoBond High Molecular Weight DNA kit from Macherey-Nagel, with the mechanical disruption of about 30 mg of lyophilized mycelium with beads for 5 min at 30 Hz. The Nanopore library was prepared with the SQK-LSK109 ligation sequencing kit, and sequencing was performed in house. The quality of the raw reads was analyzed with fastQC 0.11.9 [97] for the Illumina raw reads and pycoQC 2.5.0.23 [98] for the Nanopore raw reads (S5 Table).

In addition, the genome and transcriptome of the *S. tetrasporum* strain CBS815.71-sp3 were sequenced with Illumina technology by the Joint Genome Institute (JGI). DNA was extracted from mycelium grown on cellophane-covered potato dextrose agar medium [99]. Plate-based DNA library was prepared for Illumina sequencing on the PerkinElmer Sciclone NGS robotic liquid handling system with the Kapa Biosystems library preparation kit. Sample DNA (200 ng) was sheared into 600 bp fragments with a Covaris LE220 focused ultrasonicator. The sheared DNA fragments were size-selected by double-SPRI and the selected fragments were end-repaired, A-tailed and ligated with Illumina-compatible sequencing adaptors from IDT containing a unique molecular index barcode. RNA was extracted from the same mycelial colonies as DNA using TRIzol (Invitrogen) following manufacturer's instructions (https://assets.thermofisher.com/TFS-Assets/LSG/manuals/trizol_reagent.pdf). Plate-based RNA sample preparation was obtained with the PerkinElmer Sciclone NGS robotic liquid handling system and the Illumina TruSeq Stranded mRNA HT sample preparation kit, with the poly-A selection of mRNA according to the protocol in the Illumina user guide (https://support.illumina.com/sequencing/sequencing_kits/truseq-stranded-mrna.html), under the following conditions: 1 µg total RNA as the starting material and eight PCR cycles for library amplification. The prepared libraries were then quantified with the KAPA Biosystem next-generation sequencing library qPCR kit (Roche) and a Roche LightCycler 480 real-time PCR instrument. The quantified libraries were then multiplexed with other libraries and the pool of libraries was prepared for sequencing on the Illumina NovaSeq 6000 sequencing platform with NovaSeq XP v1.5 reagent kits, an S4 flow cell, following a 2x150 indexed run method. All raw Illumina sequence data were filtered for artifacts/process contamination. DNA reads produced by the JGI were then assembled with SPAdes v3.15.2 [100]. The obtained genome assembly is referred to as the Illumina read-based assembly of CBS815.71-sp3.

The genome of the *P. anserina* S *mat+* strain "Podan2" [55] was downloaded from the JGI MycoCosm website (https://mycocosm.jgi.doe.gov/mycocosm/home, last accessed November 15, 2018) and improved annotations [101] were downloaded from the GitHub repository https://github.com/johannessonlab/SpokBlockPaper (last accessed June 1, 2020). The genome of the OR74A *N. crassa mat A* strain [102] was downloaded from GenBank under accession number GCA_000182925.2 (last accessed May 10, 2022).

## Long read-based genome assemblies for CBS815.71 strains

*De novo* assemblies of the genomes of the CBS815.71-sp3 and CBS815.71-sp6 strains were constructed from both Illumina reads produced at Genoscope and Nanopore reads. For each genome, the raw Nanopore reads were trimmed and assembled with Canu v1.8 [103] with the option genomeSize = 35m. The assembly obtained with Canu was polished twice with Nanopore trimmed reads using first Racon v1.4.3 [104] with the options m = 8 x = 6 g = 8 w = 500 and then with medaka (https://github.com/nanoporetech/medaka) with the default settings. The assembly obtained was polished five times with the trimmed Illumina reads with Pilon

v1.24 [105]. For each round of polishing, BWA v0.7.17 [106] was used to align the Nanopore or Illumina trimmed reads with the assembly for polishing. The Illumina reads were first trimmed with Trimmomatic v0.36 [107] with the options: ILLUMINACLIP:TruSeq3-PE. fa:2:30:10 LEADING:10 TRAILING:10 SLIDINGWINDOW:5:10 MINLEN:50. The quality of the final assemblies was assessed with Quast v5.1 [108]. We checked for contamination by performing a BLAST analysis of the assembly scaffolds against the online NCBI nucleotide collection database (https://blast.ncbi.nlm.nih.gov/, last accessed March 5, 2022) and looking at the coverage of Illumina and Nanopore reads on the final assemblies. We also evaluated the completeness of the final assemblies with BUSCO v5.2.2 [109, 110] and the Sordariomyceta ortholog set (sordariomycetes_odb10, 2020-08-05, *n* = 3817 orthologous groups searched). We also ran BUSCO analysis on the Podan2 *P. anserina* assembly for comparison.

## Genome annotation

Before gene annotation, the final assemblies were formatted (contigs were renamed and ordered in descending order of size) and masked for repeats with RepeatMasker [111] and the funannotate v1.8.9 pipeline [112]. We performed gene annotation on the masked assemblies with BRAKER v2.1.6, which uses a combination of the *ab initio* gene predictors Augustus and GeneMark-ET [113]. We ran BRAKER twice, first using the BUSCO dataset of proteins sordariomycetes_odb10 [109] as evidence, and then using the RNAseq dataset as evidence. We optimized the training steps by concatenating the two homokaryotic genomes derived from the same heterokaryotic strain, CBS815.71, into a single fasta file and running BRAKER on the combined fasta file. For RNAseq mapping, we used STAR v2.7.10a [114] with the default settings. We combined the results of the two BRAKER runs with TSEBRA (the Transcript Selector for BRAKER; https://github.com/Gaius-Augustus/TSEBRA), with the option 'intron_support' set to 0.2. We selected only transcripts labeled as 't1' by the TSEBRA pipeline for downstream analyses and when BUSCO-based and RNAseq-based predictions suggested different gene structures in a single genomic locus sharing at least one coding sequence feature, we kept the transcript model with the best BRAKER evidence score or the longest cumulative coding sequence length. In rare cases, genome annotation predicted alternative gene models for homologous loci without gene-disrupting substitutions. We visualized RNAseq read mapping with the Integrative Genome Viewer (IGV) software We ran InterProScan5 v5.54–87.0 [115], TMHMM v2.0c [116], SignalP v4.1 [117] and Phobius [118] and obtained predicted protein annotations from PFAM, InterPro, EggNog, UniProtKB, MEROPS, CAZyme and GO ontology. For repeat annotation, we performed a *de novo* identification of transposable elements in the final assemblies with RepeatModeler v1.0.11 [119]. We merged the consensus sequences obtained with the RepBase library available in RepeatMasker v4.0.9 [111] and the custom library "PodoTE-1.00" built for the *Podospora anserina* species complex by [120] and the *Neurospora* library of [121] "Gioti_neurospora_repeats.renamed" (both available from the GitHub repository https://github.com/johannessonlab/SpokBlockPaper, last accessed June 1, 2020). We used RepeatMasker v4.0.9 [111] to annotate repeats. We parsed the RepeatMasker outputs and removed low-complexity and simple repeats with the parseRM_merge_interrupted.pl script from https://github.com/4ureliek/Parsing-RepeatMasker-Outputs (last accessed February 25, 2021). We retained only TEs longer than 150 bp. Overlapping TEs annotated as belonging to the same family were merged into single elements for the final annotation. We used the TE annotation suggested by [122] to sort TEs by family. We looked for signatures of RIP with the RIPper platform http://theripper.hawk.rocks (default settings; [123]). The mating-type locus was identified by tBLASTn, with the *mat+* and *mat-* idiomorphs of *P. anserina* as queries. The *SMR1*, *SMR2*, *FMR1*, *FPR1* gene models were manually curated as

previously described [124], in Artemis v18.0.2 [125]. The sequences of proteins containing a MATα_HMG domain were downloaded from the European Nucleotide Archive (ENA) database with accession numbers CDN29981 and ESA43845 for *P. anserina* and *N. crassa*, respectively. Proteins were aligned with Clustal Omega [126] and the alignments were color-coded with JALVIEW v2 [127].

## Comparative genomics and statistical analyses

For all comparative genomics analyses, we identified orthologous groups with OrthoFinder v2.3.7 [128] and studied one-to-one orthologs only. To calculate $d_S$ values of one-to-one orthologs between the CBS815.71-sp3 and CBS815.71-sp6 genome assemblies, we masked gene models for TEs. We retained only those orthologs predicted with the same number of exons and the same exon lengths in the two genomes of opposite mating types. We aligned orthologous gene coding sequences, using the codon-based approach implemented in translatorX v1.1 with default parameters [129]. We used the nucleotide alignment as input for the yn00 program in the PAML v.4.9 package [130, 131]. Pairwise Wilcoxon tests were performed in R with false discovery rate correction. We excluded the four genes located at the edges of strata to assess the differences in $d_S$ between strata.

We studied synteny between assemblies, by investigating either collinearity between one-to-one orthologs or running the genome sequence aligner nucmer from MUMmer v4.0.0rc1 [132] with default options. For the identification of small indels due to homopolymers between the CBS815.71-sp3 and CBS815.71-sp6 long-read assemblies (as known to occur in the R9 flow cell series [61]), we aligned the Nanopore-based genome assemblies of CBS815.71-sp6 and CBS815.71-sp3 and the Illumina-based genome assembly of each homokaryon with their corresponding Nanopore-based genome assembly using Minimap2 [133] with the -cx asm10 -t8 −−cs=long options. For this analysis, Illumina-based genome assembly of each homokaryon was built with SPAdes v3.13.0 (default options) from the dataset of Illumina reads produced by the Genoscope and used for polishing the Nanopore-based genome assemblies. We then used paftools from the Minimap2 [133] suite to call variants in the alignments and used bedtools v2.26.0 [134] intersect function to identify indels between the Nanopore-based assemblies also recovered in the alignment of Illumina-based versus Nanopore-based assemblies.

We identified collinearity breaks by indexing the alternative mating-type genome annotations and anchoring with single-copy homologs. We sorted the single-copy homologs based on their index in the CBS815.71-sp3 gene prediction dataset and identified collinearity breaks as changes in the index difference. Let H1, H2, H3,. . . Hn be a track of single copy homologs with indices A1, A2, A3,. . . Ai for genome A and B1, B2, B3,. . . Bj for genome B, with [i, j] ∈ N. We recorded the difference dAB between Ai and Bj ranked by the indices in A. We transposed the dAB vector per homologous contig pair and recovered the rank number in which dABr-1 was different from dABr. To reveal the underlying variants of synteny breaks we intersected the variants of the global alignment of Nanopore-based assemblies obtained with minimap2 and the call function of paftools with the CDS (coding sequence) features obtained from genome annotation. To which we added the manually predicted mating-type locus in CBS815.71-sp3. The indel variants overlapping a CDS feature and the stop-codon disrupting substitutions variants in at least one haploid assembly were displayed with the plot_synteny function of pafr (https://github.com/dwinter/pafr/).

## Genotyping of the mating-type alleles and surrounding regions

We devised a new method of DNA extraction for genotyping based on a published protocol [135]. Briefly, 1 mm$^3$ plugs were taken directly from cultures growing on agar plates and

ground with 200 μL extraction buffer, heated at 95˚C for 15 minutes, placed on ice for one minute, briefly vortexed and centrifuged at 15,900 x g for 30 seconds. We determined the mating-type allele of the offspring from a cross between CBS815.71-sp3 and CBS815.71-sp6 by PCR with the primers and conditions described in S6 and S7 Tables. Primers were designed with Primer3 software [136, 137], based on the mating-type locus region annotated in the two genome assemblies obtained for *S. tetrasporum*. We visualized amplicons by agarose-gel electrophoresis.

## Supporting information

**S1 Note. Frequency of second-division segregation of the mating-type locus under random segregation of the mating-type locus.**
(DOC)

**S1 Table. Tests of differences in growth speed and dry weight between *Schizothecium tetrasporum MAT1-1* and *MAT1-2* homokaryons and heterokaryons on different media. A.** Average and standard deviations values computed across five Petri dishes are given for each strain and medium. **B-C.** Analysis of variance (ANOVA) to test the effect of strain genotypes on growth speed (B) and dry weight (C). For growth speed, a two-way ANOVA using strain genotype and medium as factors was performed. For dry weight, a one-way ANOVA using strain genotype as a factor was performed as only one medium was used. Significant effects are indicated by stars.
(XLS)

**S2 Table. Genome assembly of the *Schizothecium tetrasporum* monokaryotic strains CBS815.71-sp3 and CBS815.71-sp6.** Table made with Quast v5.1 [1]. *BUSCO reports for highly conserved orthologs in Sordariomyceta; In the Podan2 assembly of the *Podospora anserina* S strain, BUSCO report was: C:96.8%[S:96.6%,D:0.2%],F:0.7%,M:2.5%
(XLS)

**S3 Table. List of the 15 genes in the recombination suppression region around the mating-type locus in *Schizothecium tetraporum* predicted to encode proteins that are putative transcription factors or involved in signalling pathways.** For each gene, gene ID number, location of the transcript sequence and strand as annotated in the CBS815.71-sp3 genome assembly and length, predicted PFAM domain, InterPro annotation and Gene Ontology of the encoded protein are presented. Multiple PFAM domains predicted for the same gene are shown on different rows.
(XLS)

**S4 Table. List of genes in the recombination suppression region around the mating-type locus in *Schizothecium tetraporum* having orthologues also located in the recombination suppression region around the mating-type locus in *Podospora anserina* and *Neurospora tetrasperma*.** Multiple PFAM domains predicted for the same gene are shown on different rows.
(XLS)

**S5 Table. Quality of the raw reads produced and used in this study.**
(XLS)

**S6 Table. PCR primers used in this study.** *P_MAT1-2* and *P_MAT-1-1* are primers specific to the *MAT1-2* and *MAT1-1* alleles at the mating-type locus in the *Schizothecium tetrasporum* CBS815.71 strain. The primers *P_NRR1*, *P_NRR2*, *P_NRR4*, *P_NRR5*, *P_NRR6* and *P_NRR7*

amplified sequence present at both edges of the non-recombining region around the mating-type locus.
(XLS)

**S7 Table. PCR conditions used in this study.** Different annealing temperatures were used for the primers: (i) *P_MAT1-2*, (ii) *P_MAT1-1* and (iii) *P_NRR1*, *P_NRR2*, *P_NRR3*, *P_NRR4*, *P_NRR5*, *P_NRR6*, and *P_NRR7*.
(XLS)

**S1 Fig. Genome collinearity between the long read-based assembly and the Illumina-based assembly of *Schizothecium tetrasporum* CBS815.71-sp3 strain.** Only the largest contigs (> 400kb) were plotted. Blue and red colours indicate the same and opposite strand collinearity, respectively. Genome collinearity was investigated using the genome sequence aligner nucmer from MUMmer v4.0.0rc1 [2].
(PDF)

**S2 Fig. Transposable element annotation in different genomic regions of the *Schizothecium tetrasporum* CBS815.71 strain. A.** Comparison between the whole genome (left) and the centromere (right). **B.** Comparison between the autosomes (left), the region of recombination suppression around the mating-type locus (middle) and the other recombining genomic regions on the mating-type chromosome (right).
(PDF)

**S3 Fig. Amino-acid alignment of proteins containing a MATα_HMG domain.** MAT1-1-1 proteins from *Schizothecium tetrasporum*, *Podospora anserina*, *Neurospora crassa*, and TαD1 from *S. tetrasporum* were aligned with Clustal Omega [3] and coloured with JALVIEW v2 [4]. The colour code of JALVIEW follows the Clustal X default colouring. The 80 residues of the MATα_HMG domain are underlined with a continuous line. This domain is truncated in TαD1. The domain adjacent to the MATα_HMG domain is underlined with a broken line. This domain is present in TαD1.
(PDF)

**S4 Fig. RNAseq read mapping in the mating-type locus region of *Schizothecium tetrasporum* CBS815.71-sp3 strain.** Visualization of RNAseq read mapping in the Integrative Genome Viewer (IGV) software [5]. The gene names are indicated at the bottom in black rectangles.
(PDF)

**S5 Fig. Genome-wide distribution of synonymous divergence ($d_S$) in the *Schizothecium tetrasporum* CBS815.71 strain.** $d_S$ was computed per gene between the CBS815.71-sp3 and CBS815.71-sp6 assemblies and plotted according to the CBS815.71-sp3 assembly gene position. Only genes predicted with the same number of exons and the same transcript length were considered for the analysis. The mating-type locus is indicated with a black dotted vertical line and a green triangle.
(PDF)

**S6 Fig. Delimitation of three evolutionary strata in the *Schizothecium tetrasporum* CBS815.71 strain. A.** Synonymous divergence ($d_S$) along the mating-type chromosome in the *Schizothecium tetrasporum* CBS815.71 strain and the three identified evolutionary strata 1, 2 and 3 indicated by red, green and blue rectangles respectively. The position of the mating-type locus is indicated by a red vertical line and the positions of the boundaries between strata are represented by dashed lines. The number of genes and average $d_S$ within each stratum were

computed by excluding the closest gene to each border of the stratum. $d_S$ was computed per gene between the orthologs shared between CBS815.71-sp3 and CBS815.71-sp6 assemblies and plotted according to the gene position on the mating-type chromosome of the CBS815.71-sp3 assembly. **B-C.** Variation of dS in the heterozygous region around the mating type locus when dividing the region into two segments, with the limit sliding along genes, and the difference of dS between the two segments computed for each partition and plotted according to the CBS815.71-sp3 assembly gene position. As described in the main text, two divisions were sequentially performed: (i) between 5,350,000 bp and the end of the heterozygous region to delimit strata 1 and 2; the peak of dS variation at 5,768,335 bp was considered as the boundary between strata 1 and 2 (see panel B); (ii) between the start of the putative non-recombining region and the boundary between strata 1 and 2 to delimit strata 1 and 3; the peak of dS variation at at 5,388,708 bp which was considered as the boundary between strata 1 and 3 (see panel C); Partitions with segments containing less than 10 genes were excluded from the analysis.
(PDF)

**S7 Fig. GC content variation and determination of centromeres the *Schizothecium tetrasporum* CBS815.71-sp3 assembly.** Circular plot of the CBS815.71-sp3 genome showing from top to bottom tracks: (i) Contigs from the CBS815.71-sp3 genome. Only the seven largest contigs were plotted; (ii) Gene density. Mating type locus location was shown with a green triangle; (iii) Transposable element density. Both densities were computed in 10-kb windows overlapping with 1 kb. The colour gradient shows density differences from 0% (light) to 100% (dark). (iv) Location of telomeric repeats (red bars) and putative centromere (yellow circle); (v) GC content (%) computed in 20-kb windows overlapping with 5 kb.
(PDF)

**S8 Fig. Genome-wide collinearity based on single orthologous genes between *Neurospora crassa*, *Podospora anserina* and *Schizothecium tetrasporum* genomes.** Collinearity between all contigs larger to 500 kb in each assembly. Links showed orthologous genes in the same strand (blue colour) or inverted strands (orange colour) between (i) *N. crassa* (left) and *P. anserina* (right); (ii) *P. anserina* (left) and *S. tetrasporum* (right); (iii) *N. crassa* (left) and *S. tetrasporum* (right). Genome assemblies of the *P. anserina S mat+*, *S. tetrasporum* CBS815.71-sp3 and OR74A *N. crassa* strains were used.
(PDF)

**S9 Fig. Original figure obtained with pafr showing putative deleterious mutations identified as synteny breaks near the evolutionary strata edges.** See Fig 5B for a global overview.
(PDF)

## Acknowledgments

We are grateful to J.P. Vernadet for sharing the conda environment to run the BRAKER pipeline and A. Veber for comments on a previous version of the manuscript.

## Author Contributions

**Conceptualization:** Philippe Silar, Tatiana Giraud, Fanny E. Hartmann.

**Data curation:** Nina Vittorelli, Ricardo C. Rodríguez de la Vega, Philippe Silar.

**Formal analysis:** Nina Vittorelli, Ricardo C. Rodríguez de la Vega, Christophe Lalanne, Robert Debuchy, Philippe Silar, Fanny E. Hartmann.

**Funding acquisition:** Tatiana Giraud, Fanny E. Hartmann.

**Investigation:** Nina Vittorelli, Ricardo C. Rodríguez de la Vega, Alodie Snirc, Emilie Levert, Valérie Gautier, Christophe Lalanne, Elsa De Filippo, Robert Debuchy, Philippe Silar, Tatiana Giraud, Fanny E. Hartmann.

**Methodology:** Ricardo C. Rodríguez de la Vega, Christophe Lalanne, Philippe Silar, Fanny E. Hartmann.

**Project administration:** Alodie Snirc, Tatiana Giraud.

**Resources:** Alodie Snirc, Emilie Levert, Pierre Gladieux, Sonia Guillou, Yu Zhang, Sravanthi Tejomurthula, Igor V. Grigoriev, Philippe Silar.

**Supervision:** Ricardo C. Rodríguez de la Vega, Philippe Silar, Tatiana Giraud, Fanny E. Hartmann.

**Validation:** Ricardo C. Rodríguez de la Vega, Philippe Silar, Tatiana Giraud, Fanny E. Hartmann.

**Visualization:** Nina Vittorelli, Ricardo C. Rodríguez de la Vega, Philippe Silar, Fanny E. Hartmann.

**Writing – original draft:** Nina Vittorelli, Robert Debuchy, Philippe Silar, Tatiana Giraud, Fanny E. Hartmann.

**Writing – review & editing:** Ricardo C. Rodríguez de la Vega, Robert Debuchy, Philippe Silar, Tatiana Giraud, Fanny E. Hartmann.

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
