## [Decision Letter · Decision Letter 0]

12 Aug 2022

Dear Dr. Hartmann,

Thank you very much for submitting your interesting Research Article entitled 'Stepwise recombination suppression around the mating-type locus in the fungus Schizothecium tetrasporum (Ascomycota, Sordariales)' to PLOS Genetics.

The manuscript was fully evaluated at the editorial level and by three independent peer reviewers who are all experts in the field. The reviewers appreciated the attention to an important problem, and the generation of an important data set for an interesting and unusual pseudo-homothallic fungal species.  They found the paper interesting, and the questions addressed of considerable interest.  They also appreciated the unique and broad data set generated.  This said, they did however raise several substantial concerns about the manuscript as submitted.  Based on the reviews, we will not be able to accept this version of the manuscript, but we would be willing to review a much-revised version.  We cannot, of course, promise publication at that time and a revised manuscript would be subject to review again, ideally with the same reviewers if available.

To summarize comments from reviewers to editor:

The manuscript contains interesting data about the MAT locus of a pseudo-homothallic species, but the interpretations by the authors sometimes seem to be a bit strained or stretched in several areas.  A more realistic interpretation of the data (including instances where the data do not fit a favored hypothesis) would improve this manuscript.

Key and central issues raised by the reviewers include:

1)  What is the nature of the obligate cross over event between MAT and CEN?  Is this at one fixed location and if so, how might it be generated. Alternatively, given the distance between MAT and CEN is this cross over occurring at multiple regions.  The data generated would seem to be sufficient to begin to address this, and consider possible mechanisms.

2) There seems to be some confusion about the description of dS as being high vs low in certain regions, based on comments from two of the three reviewers.

3) The model that strata of recombination is sheltering recessive alleles seems to be more of a model than an interpretation of the data presented.  What is the evidence in support of this model?  Do you observe for example obvious loss of function alleles such as premature stop codons within open reading frames?  One of the reviewers felt that the data presented in fact argued precisely against this model as the findings were not very different in comparing homokaryons and heterokaryons.  This seems a central conclusion of the paper and yet two expert reviewers came to exactly the opposite of the conclusion presented.

4. One reviewer noted:  "if an extended dikaryotic phase facilitates extension or recombination suppression, one might wonder why this is not often the case in basidiomycete mushroom species, many of which spend large parts of their life cycle in the dikaryotic phase."

5.  Another serious concern with data interpretation from one of the reviewers:

"In their analysis of evolutionary strata around the MAT locus, the authors found that there are three strata with the actual MAT locus part of the youngest evolutionary stratum. This is rather surprising and would seem to be a contradiction to the hypothesis that recombination ceased first in the MAT locus region and then suppression of recombination extended outward. The authors argue that a similar situation has been found in Cryphonectria parasitica (lines 582-583). However, in the corresponding paper by Kubisiak and Milgroom, the authors have analyzed C. parasitica strains from different populations and find recombination between the non-recombining region and the MAT locus. In fact, Kubisiak and Milgroom explicitly state that this rejects the hypothesis that this region represents a sex chromosome. Therefore, I don't see the C. parasitica case as confirmation for the authors' hypothesis. This unexpected finding would need another explanation."

If you decide to revise the manuscript for further consideration at PLOS Genetics, please aim to resubmit within the next 60 days, unless it will take extra time to address the concerns of the reviewers, in which case we would appreciate an expected resubmission date by email to plosgenetics@plos.org.

[LINK]

We are sorry that we cannot be more positive about your manuscript at this stage but do hope that you find the reviews constructive and helpful in revising the manuscript. Please do not hesitate to contact us if you have any concerns or questions.

Yours sincerely,

Joseph Heitman, MD, PhD

Academic Editor

PLOS Genetics

Kirsten Bomblies

Section Editor

PLOS Genetics

Reviewer's Responses to Questions

**Comments to the Authors:**

Reviewer #1: manuscript PGENETICS-D-22-00843

The manuscript by Vittorelli et al. describes the analysis of the mating type system of the pseudo-homothallic ascomycete Schizothecium tetrasporum. This fungus, which is a member of the Sordariales, produces mostly dikaryotic spores, and the authors showed that the dikaryotic spores carry nuclei with two opposite mating types resulting in a self-fertile dikaryotic mycelium. By isolating the (relatively rare) monokaryotic spores from five-spored asci, the authors were able to obtain homokaryotic strains with different mating types that were used for genome sequencing and analysis of the mating type (MAT) alleles, which are similar to MAT1-1 and MAT 1-2 alleles of other Sordariales. The availability of genome sequences for the mating types allowed the authors to test for recombination suppression in the region of the MAT locus. A 1.3 kb region of suppressed recombination was found extending from the MAT locus towards the telomere.

The results are interesting for researchers working on fungal mating type systems as well as sex chromosomes and related regions of suppressed recombination in general. However, there are several points where the manuscript should be improved:

1. Introduction lines 146-149 and elsewhere: The authors want to test the hypothesis that the long dikaryotic phase in the pseudo-homothallic life cycle might facilitate the extension of recombination suppression around the MAT locus. However, one caveat here would be that to check an actual correlation between pseudo-homothallism and suppression of recombination, one would also need to test more heterothallic species, where one would expect to not find this correlation. As far as I'm aware, this has not been tested in many filamentous ascomycetes, so any correlation with pseudo-homothallism needs to be viewed with caution.

Also, if an extended dikaryotic phase facilitates extension or recombination suppression, one might wonder why this is not often the case in basidiomycete mushroom species, many of which spend large parts of their life cycle in the dikaryotic phase.

2. Results and Discussion: In their analysis of evolutionary strata around the MAT locus, the authors found that there are three strata with the actual MAT locus part of the youngest evolutionary stratum. This is rather surprising and would seem to be a contradiction to the hypothesis that recombination ceased first in the MAT locus region and then suppression of recombination extended outward. The authors argue that a similar situation has been found in Cryphonectria parasitica (lines 582-583). However, in the corresponding paper by Kubisiak and Milgroom, the authors have analyzed C. parasitica strains from different populations and find recombination between the non-recombining region and the MAT locus. In fact, Kubisiak and Milgroom explicitly state that this rejects the hypothesis that this region represents a sex chromosome. Therefore, I don't see the C. parasitica case as confirmation for the authors' hypothesis. This unexpected finding would need another explanation.

3. Lines 305-308 and Figure 3C: In the text, it is stated that CBS815.71-sp6 harbors the MAT1-2 locus, but in the Figure, it's CBS815.71-sp3. Also, the nucleotide positions in Figure 3 are extremely small and difficult to read, it would be better to enlarge these and also to add the contig number for each of the strains.

4. Lines 321-323: It is stated that the transcriptional status of the pseudogene in the MAT1-2 locus is unknown. Could this be checked by using the RNA-seq data that were generated for annotation purposes for the genome sequencing process of this strain?

5. Lines 344-345 and Figure 4: The authors state that lower ds values in the center of the non-recombining region might have occurred through gene conversion. However, in Figure 4 it does not look as if the ds values are lowest in the middle (and consequently higher at the edges), rather they are low in the blue stratum, higher in red and medium in green (with higher towards the edge in the green stratum).

6. Line 349, Tables S3 and S4: The tables S3 and S4 mentioned in the text were not available in the supplementary material.

7. Introduction lines 119-126: With respect to the two mechanisms for the formation of dikaryotic spores in N. tetrasperma and P. anserina, it might be helpful to have a Figure in the Introduction showing how this works in the two fungi.

Reviewer #2: This is a thoughtful research study on a lesser known pseudo-homothallic species of fungus, using a combination of genomics, sequence analysis and experimental evidence to resolve the basis for its reproductive system. Two other species in the Sordariales have been previously studied to account for their pseudo-homothallic life cycles, Podospora anserina and Neurospora tetrasperma. This research explores a third Sordariales species, Schizothecium tetrasporum, and resolves its mating system, to feature a large region of recombination suppression associated with the MAT locus, a key cross over between this region and the centromere, and characterization of the region of suppressed recombination to suggest it arose over three stages of expansion. The research will be of interest to those investigating the basis for mating in fungi, especially convergence in evolution converting presumably heterothallic ancestors into current homothallic species.

There are considerable strengths in the genomics aspect of the research, with high quality genomes (often featuring telomere-to-telomere resolution). However, the experimental follow up is somewhat limited in its scope. For example, and while the authors acknowledge the limitation, having the set of 190 progeny is a resource that should enable a better understanding of that crossing over event. That is, is this a small ‘hot spot’, or is there sufficient distance between MAT and the centromere for this to occur at multiple sites? Second, the loss of nuclei and nature of the hyphal stages is of interest in terms of the level of stability of the homothallic state, potentially resolving the authors’ question of the maintenance of a dikaryon (e.g., do the cells really contain two nuclei as illustrated in figure 2?). Here it may have been worth exploring with microscopy on cultures looking at nucleus numbers vs. septation.

One last point is it is curious how different people interpret the same data differently. The results discussed in lines 514-550 of this study, to this reviewer at least, are very strong evidence counter to the hypothesis of sheltering deleterious alleles, since by and large there is not a huge different between the homokaryons and the heterokaryon. The third hypothesis (lines 536 onwards) is intriguing, but has the same challenge that it is difficult to see a difference between homo- and heterokaryons. Alternative explanations exist, from the perhaps unpopular concept of ‘neutral theory’ through to the potential selective advantage the ascospores cell type itself might play in the lifecycle. The data suggest that Schizothecium tetrasporum is facultatively pseudo-homothallic/heterothallic.

A few minor grammar/text considerations are as follow.

Line 36: ‘convergent association’ is unclear in meaning.

Line 50: too many adjectives, probably ‘sheltering deleterious alleles’ is better.

Line 54: ‘mutations’ plural.

Line 86: rewrite to remove double ‘associative’ ‘associated’.

Line 151: remove italics from ‘var.’,

Lines 161-163: choice of words suggests reference 21 was not inadequate, so recommend deleting ‘However’ and ‘clearly’.

Line 183: delete ‘also known as P. tetraspora’.

Line 272: delete ‘to our knowledge’.

Line 312: remove italics from ( ).

Line 342: add space ‘and 6.457’.

Line 378: remove italics from ‘spp.’.

Line 420: for consistency in style with the rest of the text, ‘Sordariales’ should be in italics in the subtitle.

Line 468: perhaps use a more descriptive subtitle.

Lines 553-562: three uses of ‘proximal’, probably all can be deleted?

Lines 563-565: worth re-writing, as duplicated ‘convergent evolution’.

Line 858: add initials to author.

Line 893: capital ‘P’ in two places.

Figure 2B: ‘apoptosis’ refers to the whole cell, rather than an organelle.

Reviewer #3: This is an intriguing manuscript. The authors investigated the breeding system of the filamentous fungus Schizothecium tetrasporum for the first time. They chose this fungus because it enables them to test their hypothesis that there is an evolutionary association (by convergent evolution) between pseudohomothallism (which involves dikaryotic nuclei of opposite MAT genotypes) and the existence of a large block of suppressed recombination around the MAT locus. The authors’ group previously proposed this hypothesis in ref 14 based on observations from two pseudohomothallic species, Podospora anserina and Neurospora tetrasperma. Now, in this study they investigated a third pseudohomothallic species, Schizothecium tetrasporum, and find that it too has a large non-recombining region around the MAT locus. Importantly, these three fungi are not closely related to each other (they are in different families).

The results are certainly interesting. The authors have done a lot of work to establish a new genetic system for studying this species: as initial work for this manuscript, they found conditions for growing and mating it, they identified a pair of parental strains that can be crossed, and they sequenced the genomes of the parental strains. The work is scientifically rigorous. The authors show suppression of recombination both in a laboratory cross (page 15) and by evolutionary analysis of synonymous sequence divergence (Fig 4), and there is good congruence between the size of the non-recombining region from these two methods. The manuscript is very well written (the Abstract and Introduction were exceptionally clear – thank you!).

==

The most intriguing aspect of this manuscript for me was the occurrence of a “single and systematic” crossing-over event between the centromere and the MAT locus, in both Schizothecium tetrasporum and Podospora anserina, leading to second division segregation (SDS) of the MAT locus in these species (as opposed to first division segregation in Neurospora tetrasperma). I am left wondering about the molecular mechanism of this single crossover, which occurs in more than 99% of meioses (Line 222). The authors do discuss this issue (Lines 493-509), but it wasn’t clear to me if nothing is known about the molecular mechanism in P. anserina (where the SDS was discovered years ago, I think).

Line 344 and Figure 4. I’m confused by the statement “the dS values were lowest in the center of the non-recombining region”. Looking at Figure 4, the opposite seems to be true because the highest dS values are in the red region (stratum 1) which is surely the center of the non-recombining region. In fact, Figure S6A shows that the median dS in this region is the highest. However, I don’t think that “lowest” on line 344 is a typo, because it is repeated on line 596/597. So I don’t understand what the authors are talking about; maybe an arrow pointing to the “low” region in Figure 4 would help.

Lines 328-400 (dS analysis section). The authors like the hypothesis that evolution selects for recombination suppression as a way of sheltering recessive deleterious alleles by linking them to a permanently heterozygous locus (MAT) (Lines 137-140, and 514-526). If this hypothesis is true, wouldn’t you predict that some of the 400 genes in the non-recombining region might contain obviously deleterious mutations (e.g. internal stop codons) in one allele, to a greater extent than in the rest of the genome? Did you look for such a pattern, and if so what did you find?

Lines 410-418. The text here is difficult to understand without looking at Figure S7, because the locations of the P_NRR markers are not shown in Figure 4. Please consider merging these two figures. Remember that many readers will not look at supplementary figures.

Lines 420-464. I didn’t really understand the purpose of this whole section about synteny of genes in the non-recombining regions (NRR) of the different species. Are you testing an hypothesis here? Some introductory sentences might help. After reading the Discussion, I think maybe that this section is connected to the “supergenes” idea on Lines 538-543, but it’s not clear. Did you test whether interspecies synteny is more conserved in the NRR than elsewhere in the genome?

Lines 507-514 are slightly unclear. Line 514 begins “An alternative hypothesis...” but it wasn’t obvious what the first hypothesis was. Reading back over lines 507-510, I wonder if the authors’ intention would be clearer if the sentence beginning on line 510 was changed to “One hypothesis could therefore be that suppression of recombination around the mating-type locus may be directly linked to the mechanism that prevents a second CO...”.

==

Graphical summary: I suggest changing “autofertile” to “self-fertile” because that is the phrase used in the text.

Lines 104-126. Does all the information in this paragraph come from reference 21, or should some other papers be cited?

Lines 121 and 125. I wasn’t sure if there is a difference between “parallel” spindles and “aligned” spindles. If they’re synonyms, please use the same word in both places. If there is a difference, please explain it and why it is important. (Also occurs on line 161).

Lines 135-137. Are non-recombining regions around the MAT locus absent in heterothallic species?

Lines 150-158. The three species are in different families. I presume that they each evolved pseudohomothallism separately, but I did not see a statement in the manuscript saying that this is the case (or that the authors assume it to be the case).

Line 307. Is there an error on this line? It says that CBS815.71-sp6 is MAT1-2, but Line 201 and Figure 3 say that it is MAT1-1.

Figure 4, please label the strata 1, 2 and 3.

Typos:

Line 54, should be “mutationS”

Line 512, should be FDS

**Have all data underlying the figures and results presented in the manuscript been provided?**

Reviewer #1: Yes

Reviewer #2: Yes

Reviewer #3: Yes

PLOS authors have the option to publish the peer review history of their article (what does this mean?). If published, this will include your full peer review and any attached files.

Reviewer #1: No

Reviewer #2: No

Reviewer #3: No

---

## [Decision Letter · Decision Letter 1]

28 Nov 2022

Dear Dr Hartmann,

Thank you very much for submitting your revised Research Article entitled 'Stepwise recombination suppression around the mating-type locus in an ascomycete fungus with self-fertile spores' to PLOS Genetics.

The manuscript was fully evaluated at the editorial level and by three independent peer reviewers including the original reviewers. The reviewers appreciated the attention to an important topic but identified some minor concerns that we ask you address in a revised manuscript.

We therefore ask you to modify the manuscript according to the review recommendations. Your revisions should address the specific points made by each reviewer.

Please let us know if you have any questions while making these revisions.  At this stage, no further review will be needed, but we wanted you to have the opportunity to address the final points raised by the reviewers through revision.

Yours sincerely,

Joseph Heitman, MD, PhD

Academic Editor

PLOS Genetics

Kirsten Bomblies

Section Editor

PLOS Genetics

Reviewer's Responses to Questions

**Comments to the Authors:**

Reviewer #1: The authors have addressed my concerns, and the revised manuscript is much clearer now.

One point for which I would like to see some more detail is the analysis of the gene disruptions etc. that the authors detected at the edges of the non-recombining regions. This is quite an interesting point since it strengthens the argument for the hypothesis proposed in Jay et al. (2022), but Figure 5A only shows a broad overview. I think that a figure (perhaps in the supplement) showing more details for each of these differences (and their distance from the borders of the evolutionary strata) would be important, also as a point of comparison for future studies.

Reviewer #2: The authors have provided a comprehensive explanation for the issues raised during the review process, and including additional analysis in the revised paper, e.g. to show the presence of mutations that would then be sheltered under their hypothesis.

Minor: during proofing, check line 171 to change "like" to "life".

Reviewer #4: The authors have addressed the concerns raised by previous reviewers and the data appear to be solid and interpretations are reasonable. Here are a few suggestions to improve the manuscript.

1. The discussion is too long and often redundant with the corresponding result sections. This makes the manuscript really long for a read.

2. The authors could also include discussions on recombination suppression observed in bipolar basidiomycetes where dikaryotic stage is long. The authors mentioned "Basidiomycete fungi are dikaryotic across most of their like cycle, but they often

have multi-allelic mating-type loci (50), allowing the loss of those with lower fitness."

3. Please check the manuscript for some spelling errors and missing periods.

4. Maybe the authors could stress that the recombination suppression is only observed in this particular heterokaryotic strain and also based on its progeny from selfing. Ideally, the phenomenon should be confirmed with additional genetically different heterokaryptic strains.

**Have all data underlying the figures and results presented in the manuscript been provided?**

Reviewer #1: Yes

Reviewer #2: Yes

Reviewer #4: Yes

PLOS authors have the option to publish the peer review history of their article (what does this mean?). If published, this will include your full peer review and any attached files.

Reviewer #1: No

Reviewer #2: No

Reviewer #4: No

---

## [Editor Report · Decision Letter 2]

17 Jan 2023

Dear Dr Hartmann,

We are pleased to inform you that your revised manuscript entitled "Stepwise recombination suppression around the mating-type locus in an ascomycete fungus with self-fertile spores" has been editorially accepted for publication in PLOS Genetics. Congratulations!

Yours sincerely,

Joseph Heitman, MD, PhD

Academic Editor

PLOS Genetics

Kirsten Bomblies

Section Editor

PLOS Genetics

Comments from the reviewers (if applicable):

**Data Deposition**

http://datadryad.org/submit?journalID=pgenetics&manu=PGENETICS-D-22-00843R2

**Press Queries**

---

## [Editor Report · Acceptance letter]

6 Feb 2023

PGENETICS-D-22-00843R2 

Stepwise recombination suppression around the mating-type locus in an ascomycete fungus with self-fertile spores 

Dear Dr Hartmann, 

We are pleased to inform you that your manuscript entitled "Stepwise recombination suppression around the mating-type locus in an ascomycete fungus with self-fertile spores" has been formally accepted for publication in PLOS Genetics! Your manuscript is now with our production department and you will be notified of the publication date in due course.

With kind regards,

Bernadett Koltai

PLOS Genetics

On behalf of:
